# Design, Implementation, and Configuration of Laser Systems for Vehicle Detection and Classification in Real Time

**DOI:** 10.3390/s21062082

**Published:** 2021-03-16

**Authors:** Nieves Gallego Ripoll, Luís Enrique Gómez Aguilera, Ferran Mocholí Belenguer, Antonio Mocholí Salcedo, Francisco José Ballester Merelo

**Affiliations:** 1Department of Electronic Engineering, ITACA Institute, Universitat Politècnica de València, 46022 Valencia, Spain; nieves.gallego.ripoll@gmail.com (N.G.R.); amocholi@eln.upv.es (A.M.S.); fballest@eln.upv.es (F.J.B.M.); 2Department of Electronic Engineering, Universidad Politécnica de Chiapas, 29082 Tuxtla Gutiérrez, Mexico; luigoaga2@gmail.com; 3Traffic Control Systems Group, ITACA Institute, Universitat Politècnica de València, 46022 Valencia, Spain

**Keywords:** classification, detection, electronics, intelligent transportation systems, laser, road safety, scanner, vehicles

## Abstract

The use of real-time vehicle detection and classification systems is essential for the accurate management of traffic and road infrastructure. Over time, diverse systems have been proposed for it, such as the widely known magnetic loops or microwave radars. However, these types of sensors do not offer all the information currently required for exhaustive and comprehensive traffic control. Thus, this paper presents the design, implementation, and configuration of laser systems to obtain 3D profiles of vehicles, which collect more precise information about the state of the roads. Nevertheless, to obtain reliable information on vehicle traffic by means of these systems, it is fundamental to correctly carry out a series of preliminary steps: choose the most suitable type of laser, select its configuration properly, determine the optimal location, and process the information provided accurately. Therefore, this paper details a series of criteria to help make these crucial and difficult decisions. Furthermore, following these guidelines, a complete laser system implemented for vehicle detection and classification is presented as result, which is characterized by its versatility and the ability to control up to four lanes in real time.

## 1. Introduction

In recent years, the transport and logistics sector has undergone a strong transformation. Technologies present in our day to day such as mobile applications, internet, big data, or artificial intelligence have countless applications in this field which, despite having already begun to be developed, may change drastically the way in which travelers and goods are transported in the near future.

The main world capitals are places with a high level of accumulation and concentration of economic activities. However, the larger a city is, the greater its complexity and potential for disruption, and unfortunately, the number of inhabitants in cities is increasing steadily. Therefore, the future sustainable development of cities will be determined by the proper management of urban growth, especially in low- and middle-income countries [1].

Traffic congestion is one of the main problems facing cities today. In fact, it continues growing in urban and interurban areas. In consequence, more and more cities have realized that investment priority should be given to the intensive use of smart systems and telecommunications applied to traffic management. In effect, the so-called Intelligent Transport Systems (ITS) are becoming an efficient support for the citizen and for public institutions in the attempt to alleviate the problems of congestion of urban and interurban transport, not only helping to improve their mobility but also making it more sustainable. To achieve these objectives, greater control over vehicles is required [2], which is why real-time vehicle detection and classification systems have become a primary need within Intelligent Transportation Systems (ITS)—hence the importance of this study.

Transport supply and demand have a reciprocal but asymmetric relation. While a realized transport demand cannot take place without a corresponding level of transport supply, a transport supply can exist without a corresponding transport demand. This effect is compounded when designing traffic plans for cities, which is an arduous task that is usually carried out by the entities in charge of traffic management of the respective cities. For example, planning for peak capacity leaves the system highly under-used during off-peak hours, but planning for an average capacity leads to congestion during peak hours. 

During the last decade, there has been a significant development of intelligent information technologies for vehicle route management [3], increasing processing capabilities and developing better planning systems and techniques [4]. This constant development led to the continuous appearance of new designs that offered greater benefits than their predecessor. However, the reliability of these new devices was generally directly proportional to their price, which is why, due to their low maintenance costs, their widely known technology, and their great durability, magnetic loops continue to predominate in road infrastructure around the world [5].

Magnetic loops were massively introduced in the early 1960s and are the most common sensor on roads around the world. In fact, they have become the benchmark road sensor. This is mainly because these detectors have proven to be very efficient over the years. Apart from their main application aimed at counting and classifying vehicles, which includes buses, trucks, cars and motorcycles [6], these are also used for speed measurements, access control, estimation of traffic density, shadow tolls, and even for two-way communication between vehicles and road infrastructure [7].

The operation of these sensors is based on the variation of inductance that is registered in the loops during the passage of the vehicles over them. These detectors are buried in the pavement and strategically installed to form part of an oscillator circuit, which is interconnected with an electronic unit that is usually located on the nearest sidewalk. In this way, when a vehicle or any ferromagnetic object crosses the magnetic field generated by these, there is a decrease in the global magnetic field due to the parasitic currents induced in the vehicles (eddy currents), which in turn causes a decrease in coil inductance. 

However, the potential of magnetic loops is limited. The existing configurations of single loops do not allow obtaining such important parameters as speed, length, or direction of movement of vehicles. This is the reason why the vast majority of them are usually only used in urban environments to measure traffic flow and count vehicles. To estimate the speed of a vehicle, its length, or its direction of movement, two individual loops are required. For this reason, usually two magnetic loops spaced a certain distance are installed per lane; this is commonly known as a “dual system”, and it is a configuration that can be found on practically all the highways on the planet and on the busiest roads in the main cities.

Nevertheless, even these dual systems do not provide all the information necessary to carry out highly detailed traffic control. For example, magnetic loops cannot obtain specific characteristics such as the height of vehicles or the existence of trailers [8]. The classification of vehicles offered by these is limited, and the margins of error made in calculating the speed and length of these are usually high (around 20%). In addition, the use of intrusive systems complicates the installation, removal, and repair of the sensors, since these tasks cannot be performed without interrupting traffic. Thus, recent research in this field has shown that other technologies that are widespread around the world but little exploited in the ITS sector could improve the results offered by these traditional systems [9].

Over-roadway sensors, such as laser systems, do not require the installation of the sensor directly onto, into, or below the road surface, as they are habitually mounted over the roadways or to the side of them. Video image processor, microwave radar, active and passive infrared, ultrasonic, and passive acoustic array are technologies applied to these types of sensors. In this sense, it might be thought that closed-circuit television (CCTV) could be a good alternative to magnetic loops and laser systems, or even the simplest and optimal solution for vehicle monitoring. CCTV can replace several in-roadway inductive loops and provide detection of vehicles across several lanes, which implies lower maintenance costs. In addition, some CCTV systems process data from more than one camera and further expand the area over which data are collected. In this way, these systems classify vehicles by length and report on vehicle presence, flow rate, occupancy, and speed, which are the same parameters offered by laser technology.

However, some disadvantages of the video image processor compared to laser technology include its vulnerability to viewing obstructions (tall vehicles such as trucks can block the line of sight), inclement weather, shadows (vehicle projection into adjacent lanes), day-to-night transition, vehicle/road contrast, and dirt on the camera lens (which can affect performance). To solve these problems, powerful software with algorithms is required, which is not usually included in these systems. In addition, some models are susceptible to camera motion caused by strong winds. Furthermore, a video image processor arrangement is generally cost effective only if many detection zones are required within the field of view of the camera and they cannot obtain the height of the vehicles nor the 3D profile of these—hence the emphasis on laser proposals.

These sensors are based on light detection and ranging technology (LIDAR) for vehicle detection and classification [10,11]. Their operation is straightforward. Firstly, the laser emits multiple pulses of infrared light, which bounce off the vehicle and return to it. Secondly, the system measures the phase shift (angular or temporal, depending on the model) between the reflected and the emitted beams. Finally, the information is processed to obtain all kinds of characteristics of the vehicle [12,13,14]. Its mode of operation is shown in Figure 1.

They are usually located on or on the side of the road, which does not directly interfere with traffic during their maintenance and commissioning [15]. Figure 2 shows the installation of a typical laser system commonly placed on variable messaging panels and an example of a laser gun with the same technology, which is usually used on the sides of the road. The most common use of both settings is usually to fine vehicles that exceed the speed limit, but this paper will show that by developing the appropriate hardware and software, the potential of this technology can be much greater.

Most of these systems perform a sweep perpendicular to the road, cutting the vehicles transversely, which allows counting and classifying them by analyzing their silhouette. However, these detectors also allow obtaining many other characteristics of the vehicle such as height, lane position, presence of tow bar, length, width, speed, opposite direction, and traffic flow.

The main bottleneck for LIDAR technology today is processing the large amount of data acquired with laser scanning devices. Throughout the years, many tools have been developed for point cloud data processing. Several studies have been conducted on infrared laser-based sensor systems as researchers began to investigate the potential of these sensors for traffic counting, classification, and speed estimating [15,16,17,18,19]. The theory behind this research was that laser receivers could detect the absence and presence of single or dual laser beams to determine the presence of vehicles. However, in all cases, there were great limitations regarding the number of sensors to use, the lanes to monitor, the number of vehicles to be differentiated, or the maximum speed at which these could move. For example, these systems were generally limited to speeds below 80 km/h [16] and could monitor a maximum of two lanes, which was totally insufficient for exhaustive, detailed, and comprehensive traffic control. 

Nevertheless, these tedious and difficult processing tasks are tending to disappear or be minimized with the appearance of machine learning algorithms. These tools allow not only the development of advanced, efficient, and intelligent processing but also the interpretation of data. Accordingly, this paper aims to provide a solution to all these drawbacks and, therefore, to offer a versatile and multifunctional system optimized for use in urban and interurban environments. However, it should be noted that due to the complexity of the statistical analysis performed, the mathematical method applied, and the algorithms developed, everything related to machine learning will be described in detail in another paper. This one simply intends to present the implemented system and offer a step-by-step guide on how to design a laser system for road applications.

## 2. Materials and Methods

### 2.1. Methodology

LIDAR technology is becoming increasingly popular in the road sector around the world. This trend can be seen in the growing number of agencies acquiring LIDAR scanners and contracting LIDAR services [19]. The primary factors behind this trend are that, on the one hand, surveyors, engineers, and technicians are becoming more educated about and increasingly open to LIDAR and, on the other hand, LIDAR is potentially more cost-effective than traditional technologies. 

Although this may be a bit less economical, it offers much more information, precision, and durability than other systems, which is compensated and amortized over time. Therefore, as the realization of a new system for the detection, information extraction, and classification of vehicles in real time through laser technology requires a certain methodology for its development, this paper aims to be a guide to understand and know how to properly configure road sensor systems based on LIDAR technology. Figure 3 shows the entire procedure to configure a laser system correctly in detail.

To standardize this system and facilitate the future implementation of this technology in the road sector, a series of technical specifications have been established:➢It must be able to function properly regardless of its placement and laser sensor used.➢It must provide an intuitive and user-friendly graphical interface.➢Calibration must be available whenever required.➢It must be as unaffected as possible by inclement weather such as rain, snow, and fog.➢It must control up to four lanes in real time.➢It must discriminate up to 8 + 1 vehicle categories according to the classification of the German TLS standard defined by the Federal Institute for Highway Research (Bundesanstalt für Straßenwesen) [20], which has been taken as a reference in much of the world. It includes 8 vehicle categories (motorcycles, cars, cars with trailer, van, trucks, trucks with trailer, bus, and articulated vehicles) and the category “unknown”. Hence the 8 + 1 nomenclature.➢It must be able to provide information on vehicle detection, classification, lane position, presence of tow bar, length, width, height, speed, opposite direction, and traffic flow.

With the number of cars clogging roads around the world expected to double in the coming decades, new ways of responding to crashes, controlling traffic lights, and creating diversions will be needed to keep traffic moving. Currently, there are many companies that offer laser equipment to solve these issues [21], but the adaptation and use of these laser devices for the detection and classification of vehicles is not usually an easy task, since it depends on the operating mode of each particular laser. In most cases, many non-trivial adjustments are required to ensure proper system operation [16], which hinders and slows down the massive deployment of this technology. Hence, the goal of implementing a versatile system that works properly regardless of the laser sensor used.

### 2.2. Choice of Laser Device

The choice between the commercial units offered today in the market has been made based on their presence in current ITS systems and in the bibliography. Thus, three of the equipment most used by different authors [22,23,24,25] and a multisensor system were selected for analysis.

➢LMS 221-30206 of the company SICK AG. This is very widespread equipment both in ITS applications for vehicle detection and in robotics to aid navigation and obstacle detection.➢AccuRange from the company Acuity. These are actually two independent units. AccuRange 4000 is the laser scanner itself and AccuRange Line Scanner is made up of the motor and the rotating mirror that make scanning possible.➢AutoSense 800 from the company OSI LaserScan. This equipment, widely used in ITS applications, is characterized by higher signal power and emitted wavelength.➢TT 293 of the ASIM company. This is a multisensor system composed of a Doppler radar in charge of making speed measurements, ultrasound sensors to detect and classify vehicles, and a passive infrared sensor (PIR) sensor.

Table 1 shows the key parameters regarding the commercial equipment studied, which were obtained directly from their corresponding datasheet. For a better understanding, the specifications have been grouped under four concepts: general, communications, scanner, and other. In this way, while the scanner category includes the most relevant data depending on the application, the rest are influential in determining the infrastructure, location, and budget.

After the study of the previous different commercial units, we selected the one that was best suited to the application to be developed. Thus, although at first glance, it might seem that the Autosense 800 has the best performance in terms of speed, it has a low scanning angle, which is insufficient for urban roads with up to 4 lanes. In view of this information, the choice was between the following:➢LMS 221-30206: high-performance sensor capable of sweeping on a multilane road. The unit emits a laser beam through a diode whose diffuse reflection is received by a photodiode. To perform the sweep, it uses a rotating mirror powered by a stepper motor. This unit also offers the ability to select the scanning frequency, the angular resolution, the response time, and even the transfer rate.➢AccuRange 4000: the main difference with respect to the previous one is that this works with a direct current (DC) motor, which together with an encoder provides information about the position in which the measurement is made.

Thus, although both models had similar characteristics and could be used for our application, the scanning frequency and the high-speed option available in the case of the LMS led to us choosing this one. However, this choice does not condition our design, since as previously commented, one of the goals of this design is precisely that the accurate operation of the system does not depend on the laser used. For this reason, a smart user interface so that the sweep can be carried out independently of the laser model has been implemented. Furthermore, to minimize any possible dependence on the sensor equipment, a printed circuit board (PCB) has also been designed, which allows adaptation of the signals received from the sensor to a standard frame.

### 2.3. Operating Mode

A laser system is based on an area detector, which is a device that works by measuring the flight time of a laser beam, as shown in Figure 1. A pulsed laser diode emits a beam toward the detection zone, in this case, the road, and its diffuse reflection is recorded by the receiver. According to the nomenclature used in Figure 1, forward transit time corresponds to the time it takes for the laser beam to reach the vehicle, while backward transit time corresponds to the time it takes for the laser beam to be captured again by the laser system once it bounces off the vehicle. In telecommunications, the sum of these parameters corresponds to the round-trip time: the amount of time it takes for a signal to be sent plus the amount of time it takes for an acknowledgment of that signal to be received. In this way, as these are directly proportional to the distance between the sensor and the vehicle, the distance is determined by measuring the time between sending the beam and detecting the reflection. Thus, knowing the distance to the road in the absence of vehicles, the height of vehicles that pass through the detection zone of the system can be obtained. Figure 4 shows the detection zone of a conventional two-lane highway. The scanning angle α means the total spread at which the scan is carried out, which is determined by the width of the road and the height at which the sensor is installed. The result of this process is a dataset in the form of a point cloud containing the position of every detection point.

In this sense, it might seem that the electronic unit directly measures heights, but this is not entirely true, since this actually measures the distance from its position to the surface on which it is focused. To obtain the height of the vehicles, the electronic unit uses geometric calculations. Figure 5 shows a schematic of how height measurement is performed.

First, an initial scan in the absence of vehicles is made, which generates a reference or offset vector. This reference vector is basically the laser distance to each point on the road when there are no vehicles (measure B in Figure 5). However, each of these distances is measured with a determined scanning angle. Thus, it is necessary to know the angle that corresponds to each of the measurements when calculating the height values. In this way, the values of the heights of each of the points (YB) can be obtained by multiplying the value of B by the cosine of the corresponding angle α. Under ideal conditions, all the values of this vector should be the same, but sometimes, they may differ due to irregularities in the asphalt or static objects located on the road. Nevertheless, these vectors are stored in the equipment configuration and can be measured again if required.

In the case of vehicle detection, the operations performed are similar. When there are vehicles on the road, the measured value corresponds to A in Figure 5, and calculating the angle effect yields YA. Then, to obtain the actual height of the detected vehicle, both vectors are subtracted.

The measured distances are encoded in two bytes. In the standard configuration, that is, using measurement ranges of up to 8 m, only bits in the range 0–12 are used to represent the distance (bits 13–15 are unused). In our tests in real environments, the unit was located 6 m above the road; therefore, the configuration used 13 bits (0–12) to encode the distance information. These 13 bits allow values of up to 2^13^ − 1 = 8191 to be represented, which when working in millimeter resolution becomes 8.191 mm. However, since we aim for our system to not depend on the chosen measurement equipment and to be totally adaptive, in other designs with different requirements and needs, depending on the range of measurements, the numbers of bits shown in Table 2 should be considered. The implemented software also takes this information into account, which selects the number of bits according to the maximum measured value.

### 2.4. System Implementation

After acquiring some knowledge about the equipment and its operation, an electronic circuit was designed. The purpose of this was to serve as an interface between any commercial laser scanner sensor unit and a computer. This was basically made up of three large blocks: the converter stage, the microprocessor, and the power supply.

The converter stage is responsible for conditioning the communications between the laser sensor and a personal computer. In our case, the chosen LMS unit transmits the signals using the international standard RS422, but given the generalization of the USB standard in current computers, the output of the PCB to the computer was modified, which allows greater flexibility and adaptability. In this stage, the appropriate connectors and the electronic circuitry necessary for communication between the different elements of the system were also designed and built.

A microprocessor is used to acquire and process the signal prior to sending it to the computer. To avoid sending irrelevant information to this (when there are no vehicles on the road), it only sends signals when these are detected. In this way, during long periods in which there is practically no traffic, such as at night, the information channel is not saturated with irrelevant data. However, the microprocessor is also in charge of performing the following functions: adaptation and prior conditioning of the system depending on the selected laser, configuration of the laser scanner characteristics, and normalization of the laser plot (correction of the angle effect, detection and elimination of static elements on the road and sending information to the computer).

The selected microprocessor was a digital signal controller, since it has the same functionalities as a digital signal processor (DSP) but with greater benefits [26]. This microprocessor can be programmed normally in the C programming language, which offers two important advantages. The first and main advantage is that this allows modification, verification, and debugging of the design without affecting the hardware architecture, and the second one is that this facilitates the reuse of the microprocessor at the end of the project. Therefore, given its versatility, a connector was also included in the PCB design for programming the digital signal controller (DSC) if necessary.

Regarding the power supply, it should be noted first that the components of the PCB work with voltages of 5 V. Thus, an adaptation stage was necessary. This adaptation stage was made up of various discrete components, which protects the circuit against voltage variations, electric shocks, and existing noise and filters the signal. In addition, for a quick visual check of the accurate operation of the system, several LED indicators related to the different stages were added, which facilitates the review and detection of errors. Figure 6 and Figure 7 show the different possible communications on the board and the PCB designed with each of its components.

As the system was designed to be used independently of the laser sensor, it was necessary to consider the different communication standards. This is why the PCB has various electrical connectors: DB9, both male and female for serial communications, USB-B connectors, and RJ-11 connectors for programming. Therefore, the defined architecture and specifications can accommodate a wide variety of combinations. In this way, the information recorded by the equipment is sent through an RS-422 interface to the board, where it is transformed at 5 V, and once in the PCB and depending on different combinations of jumpers, the data flow can be as follows:➢Equipment—Serial communication: the data are sent to the TTL-RS232 converter and directly to the computer for direct processing there.➢Equipment—DSC communication: the data are sent to the DSC for preprocessing, converted to RS-232 levels, and finally sent to the computer through its serial port.➢Equipment—DSC communication: after the data are preprocessed in the DSC and transformed to RS-232 levels and then the USB, they are sent to the computer through that port.

Once the hardware architecture specification stage was successfully completed, which implied the study of the principle of operation of laser scanner sensors, the establishment of the communication stages between the sensor equipment and the computer, and the design of the necessary hardware, the final PCB was placed inside a box as shown in Figure 8.

Once the hardware needed to acquire the signals correctly was designed and efficiently implemented, the software in charge of the processing and treatment of all the data was added. This is represented in Figure 9 in a simplified way by means of a block diagram. The selected digital signal controller (DSC) is a dsPIC30F4011 from The Microchip Company, and therefore, the development environment MPLAB IDE version 7.6 from the same company was used, since it allows editing, compiling, and debugging of the designed programs. The firmware programming was performed in the programming language C.

## 3. Results

Through the designed hardware system and its corresponding software, we were able to implement a robust system that met the requirements established at the beginning of the paper and achieved the specifications shown in Table 3. Consequently, after having carried out both hardware and software design and proceeding with its implementation, the next step was to proceed with the installation of the system in real environments, evaluate its operation, and validate it by analyzing the results.

The site selected for installation was the junction of the Carrera Malilla and Isla Formentera street (Valencia 46026) in the southern part of the city. After the urban growth that formed the south round, this became the main exit from the neighborhood, which ensured a high flow of vehicles. Figure 10 shows the satellite view of the area, where the star indicates the location of the pole on which the scanner laser sensor was installed.

In the detection zone, two lanes were controlled in each direction as Figure 11 shows. Direction A referred to vehicles entering the neighborhood, while address B represented vehicles leaving it. In addition, as the zone was an intersection between a main road (Carrera Malilla) and a secondary one (Isla Formentera), the green phases of the traffic light were longer on the main road, giving priority to this.

The installation of the laser scanner sensor was carried out by the maintenance teams of the Valencia City Council and our research group, who cooperated in the research. The installation required the assembly of a special pole on which the unit was placed next to a video camera and a cabinet at the side of the road. The camera allowed us to obtain information from the detection area, which was used later to verify the results of the system and to store the images of the detected and classified vehicles in a database.

Experiments and tests were carried out for more than a year over three different periods. Early in the morning (between 8:00 and 11:30 am, when the light conditions on the road provided low light intensity), at midday (between 12:00 and 2:00 p.m., when the light intensity increased as it fell directly on the road) and at dusk (between 9:00 and 10:00 p.m., when the atmospheric light intensity was practically nil). In this way, this long period of time made it possible to carry out tests at all times of the year, which verified that the detection and classification were not affected in any case by the season of the year, luminosity or temperature differences. This verification was carried out by analyzing different captures and comparing them with the corresponding images taken by the camera, which ensured that the classification was accurate. Figure 12 shows some of the 3D profiles registered by the system after the passage of different vehicles on the laser sensor.

After obtaining the 3D profiles, vehicle classification was carried out by the real-time software system based on 33 parameters [27,28], which will be presented in detail in an upcoming paper. Nevertheless, the predictive parameters used by the algorithms are shown in Table 4.

After defining certain guidelines and patterns based on these statistical analyzes, a classification was achieved using tree techniques according to the aforementioned German TLS standard (8 + 1 vehicle categories). Figure 13 shows the classification offered by the software. Furthermore, the system was designed so that these results can be sent to traffic management centers in real time using any type of communication technology: fiber optic cable, coaxial cable, GPRS, GSM, WiFi, or WiMAX. In fact, during field tests in Malilla, WiMAX technology was used to communicate with the Valencia City Council traffic control room from the system site. Figure 14 shows the graphical user interface of our system at the described location and the Appendix A, which describes how it works in real time.

Finally, to corroborate the proper functioning of our equipment, the system was also used in other scenarios with different requirements and lasers. The Appendix A shows the operation of our system in one of the most important avenues in the city (Avenida Blasco Ibáñez) as part of the European project TRACKSS of the VI Marco program, whose objective was the implementation of innovative technologies for advanced roads. On this occasion, the laser used was not the LMS 221-30206, but the AccuRange 4000 model, which corroborated the correct operation of the designed system regardless of the laser sensor used.

## 4. Discussion

Currently, cities are subject to numerous factors that complicate their proper functioning. These problems range from traffic congestion and energy consumption to noise pollution, including pollution from vehicles and even the dehumanization of some urban spaces. Among all these factors, attempts are made through innovative methods to find solutions that can facilitate people’s daily life, their quality of life, and their journeys. However, traffic congestion on urban road networks has increased substantially, and delays are becoming more frequent. Congestion not only degrades the performance of the traffic systems themselves but also decreases the productivity of people and companies. Due to this, the quality of life of people residing in cities worsens.

Moreover, congestion losses in cities have recently been reaching exorbitant numbers of billions of euros every year. Specifically, traffic jams cost developed countries approximately 2% of their gross domestic product (GDP) annually, according to estimates by the Organization for Economic Cooperation and Development (OECD).

Intelligent transport systems can address this problem. These offer opportunities to manage travel demand and can help reduce the need for new infrastructure. ITS is the integration of information and communications technology with transport infrastructure, vehicles, and users. It enables information to be collected and shared to help people make more informed travel choices, make journeys more efficient, and help reduce the impact of transport on the environment. Traffic control and management systems have proven to improve the mobility of users and simultaneously guarantee road safety for all. Consequently, our research group managed to develop a new intelligent system for the management and optimization of urban traffic, which is an innovative tool in line with the global trends of the Internet of Things (IoT) and artificial intelligence. In addition, the proposed system was designed to be installed in both urban and interurban environments, which directly impacts society and has a series of immediate benefits: redirection of traffic based on congestion, reduction in vehicle travel time, access control to restricted areas for environmental and antiterrorist purposes, verification of road regulations, payment of shadow tolls, and reduction in accidents.

The system was tested in different real settings, and the results obtained in the detection and classification evaluation demonstrate that the data are accurate and reliable. The detection, classification, and information extraction, capable of covering up to four one-lane lanes, offered the following key features:➢Detection ratio: 97.894%.➢Precision: 99.694% (ratio between the correct classifications of a class and the total number of vehicles classified in that category).➢Accuracy: 97.60% (ratio of correct classifications to all classifications).➢Margin of error in the calculation of the speed and length of the vehicles: less than 3%.

These were excellent values that confirm the precise and adequate definition of the functional requirements and the proper implementation of the system. The aforementioned magnetic loops provide margins of error in the calculation of the speed and length of the vehicles of around 20%. In CCTV systems, this is usually around 5%. Even the widely used microwave radars provide higher margins than our system provides (4%). In addition, the implemented system provides versatility, which not all current systems offer.

## 5. Conclusions

Intelligent transportation systems play a key role in supporting socioeconomic activities all over the world, and especially, enhancing the activity and attractiveness of urban areas requires a drastic improvement in road infrastructure to provide users with better services, high reliability, and low emissions. The vast majority of countries, both developed and underdeveloped, are currently facing big challenges because of the extraordinarily fast motorization over the past years. Inadequate infrastructure relative to the number of vehicles, coupled with largely obsolete vehicle fleets and outdated technical inspection systems, leave many of these countries with high accident statistics and poor environmental records.

These high levels of urbanization are forcing governments to look for innovative solutions in this fight against congestion, pollution, and collisions. Currently, the private car and the moped constitute the main means of movement, even for short distances. However, when planning and executing urban and road policies, the specificities of each place are not always considered. In this sense, it is crucial to consider that each city is different both in socioeconomic terms and in the type of dominant activity. Likewise, the modes of travel and the type of transport associated with them differ from a city linked to livestock compared to another focused on industrial production or tourism. Consequently, each city requires a particular solution, according to the needs of their inhabitants. This is where intelligent transport systems and especially the system presented in this paper bring great benefits, since they can adapt to different situations and provide accurate information regardless of the scenario, which provides information in real time to the control and management centers so that they can design the appropriate traffic plans and adapt to the transport needs.

Therefore, a research that summarizes the state of the art of LIDAR technology, defines a series of steps for the correct configuration of laser equipment for road environments, and presents a complete system whose massive deployment would help mitigate the aforementioned problems has been presented. The most relevant applications related to monitoring and inventory transport infrastructures have also been discussed. Furthermore, different commercial LIDAR systems have been described and compared with different technologies to offer a broad scope of the available sensors and tools to remote monitoring infrastructures.

In this sense, leaving aside the techniques of signal processing, which will be explained in detail in another paper, it can be appreciated that the main contribution of this system are the following:➢The optimization of a technology (widely extended but little used in the ITS sector) to be used in the road sector.➢The addition of functionalities to a sensor that nowadays is basically used only to measure the speed of vehicles and fine them if necessary.➢The homogenization of a technology that currently requires specific software and hardware depending on the laser used, which hinders its expansion.➢The design of hardware and software that allows vehicles to be monitored independently of the laser used, whose possibilities are innumerable.➢The ability to obtain the height of the vehicles, which is very useful for detecting vehicles that are too high. These large vehicles can damage infrastructure and even cause accidents in tunnels and bridges.➢The implementation of a laser system that works with 3D magnetic profiles, performs very powerful signal processing, and has proven to work flawlessly under different conditions in the city of Valencia.

According with study realized by The United Nations Population Fund (UNFPA) [29], “In the years to come, more than half of the world’s population will be living in urban areas. By 2030, towns and cities will be home to almost 5 billion people”. This radical rise of population that lives on cities is forcing us to optimize some aspects of cities, but there are some limitations with traffic, as the city layout is complex and cannot be changed drastically. This is the reason why our research group has opted for such research, having managed to implement an adaptive laser system for road applications from scratch, as well as a step guide for future implementations. In fact, it is currently still in operation in the city of Valencia, offering information of great interest to the city’s traffic control room.

## Figures and Tables

**Figure 1 sensors-21-02082-f001:**
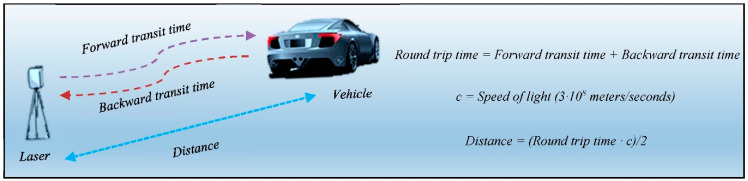
Operation of a laser sensor.

**Figure 2 sensors-21-02082-f002:**
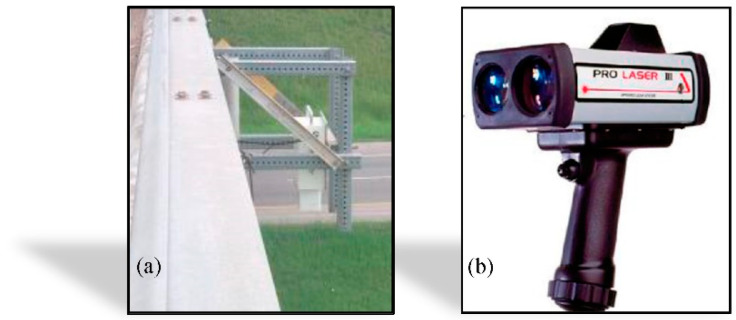
Laser technology. (**a**) Installation on bridges. (**b**) Laser gun.

**Figure 3 sensors-21-02082-f003:**
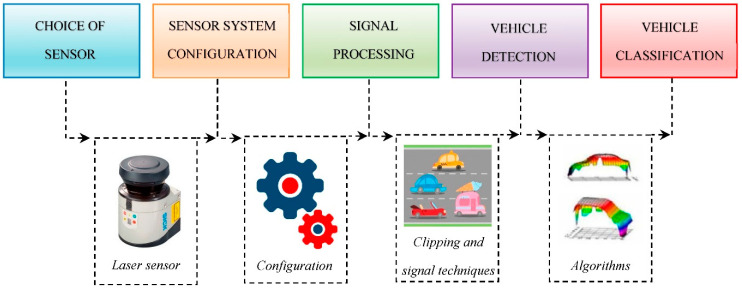
Procedure to configure a laser system correctly.

**Figure 4 sensors-21-02082-f004:**
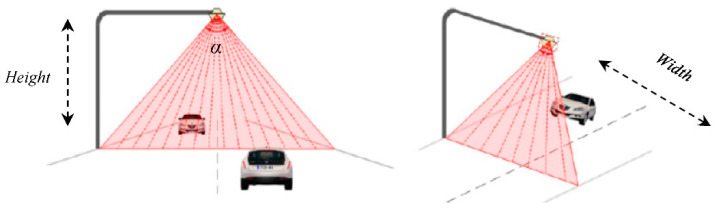
Principle of operation.

**Figure 5 sensors-21-02082-f005:**
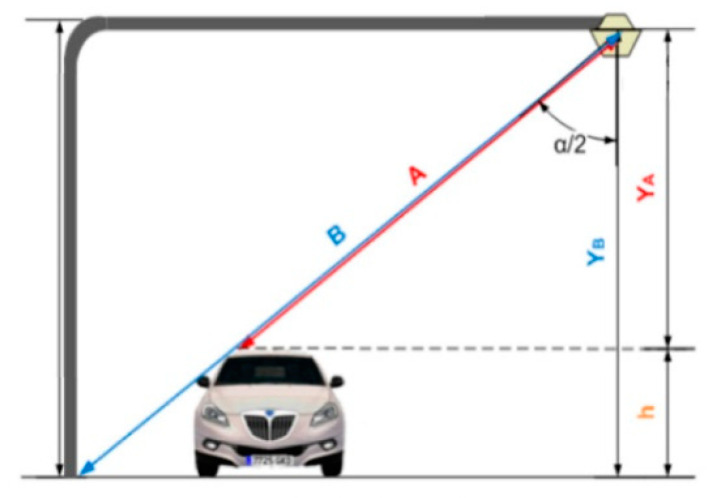
Height measurement.

**Figure 6 sensors-21-02082-f006:**
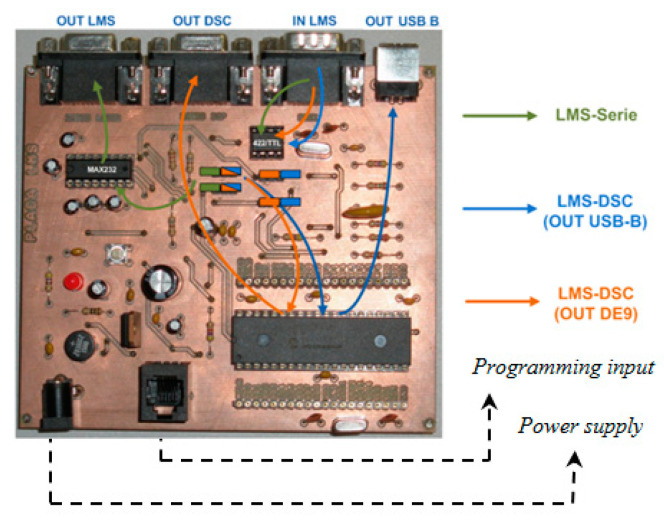
Printed circuit board (PCB) communications.

**Figure 7 sensors-21-02082-f007:**
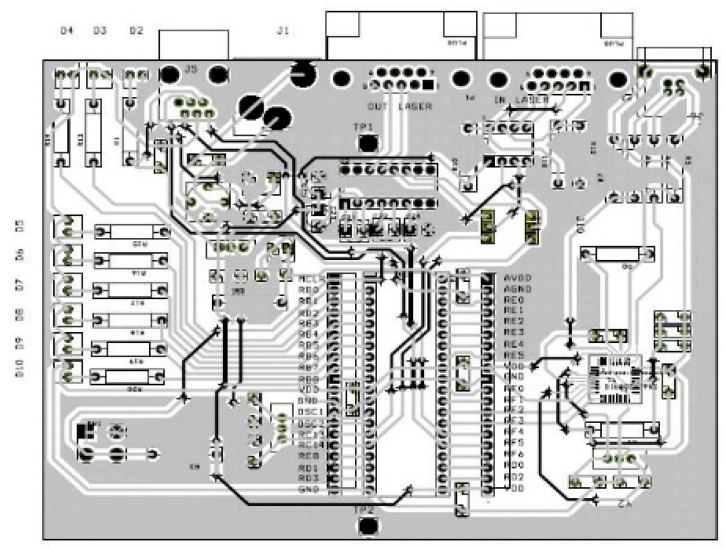
PCB layout.

**Figure 8 sensors-21-02082-f008:**
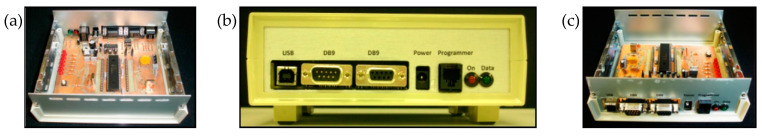
Aspects of the complete system. (**a**) Top view, (**b**) Front view, (**c**) Back view.

**Figure 9 sensors-21-02082-f009:**
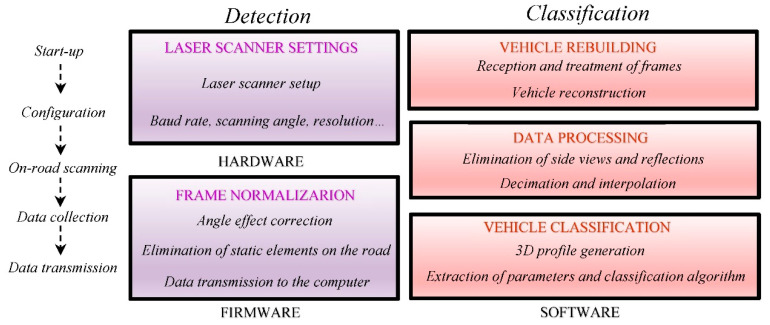
General system architecture.

**Figure 10 sensors-21-02082-f010:**
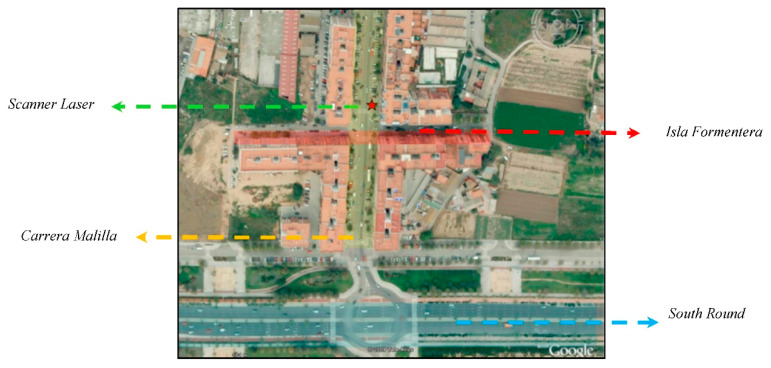
Satellite view of the area where the laser sensor was installed.

**Figure 11 sensors-21-02082-f011:**
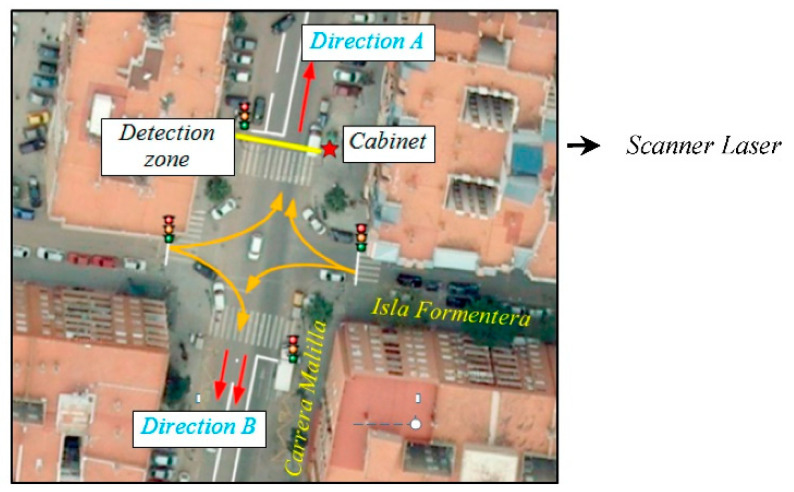
Intersection under study.

**Figure 12 sensors-21-02082-f012:**
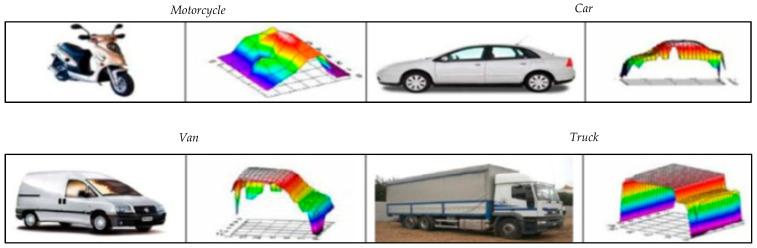
Some 3D profiles extracted from different vehicles.

**Figure 13 sensors-21-02082-f013:**
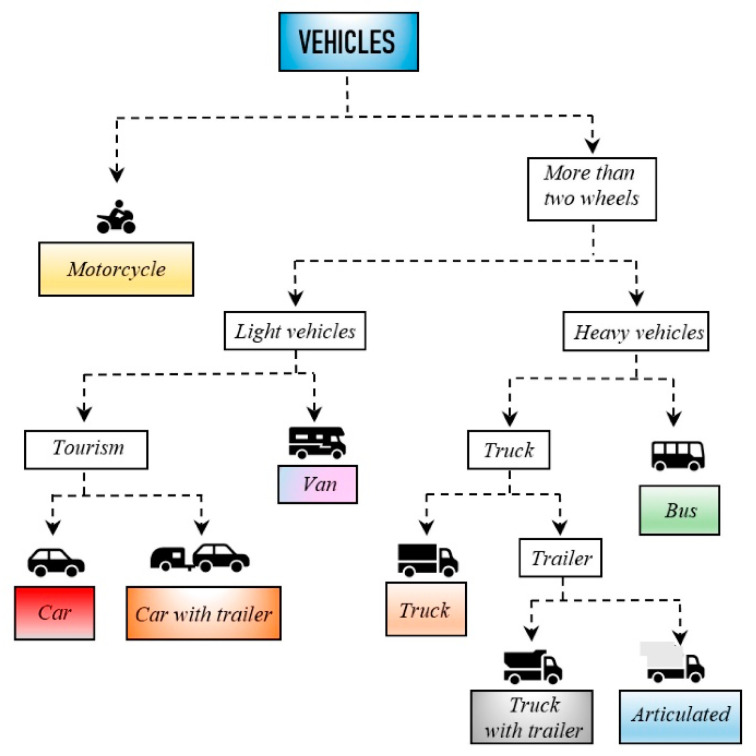
Tree diagram of the discrimination performed.

**Figure 14 sensors-21-02082-f014:**
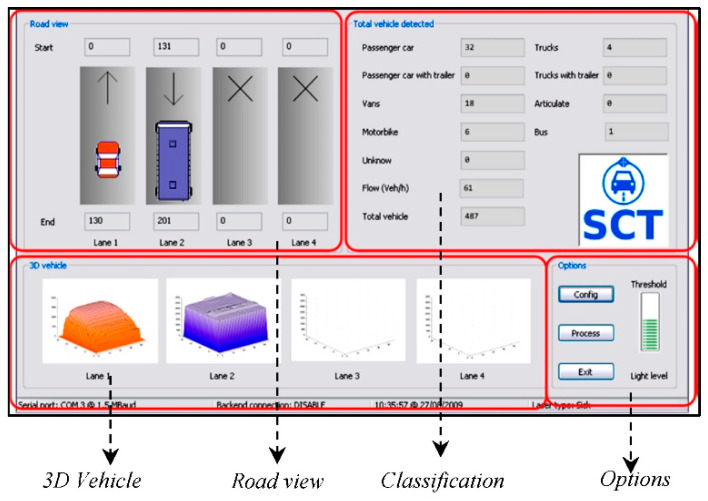
Operation of our system at the described location.

**Table 1 sensors-21-02082-t001:** Comparison between commercial units.

	Company Model	Sick LMS 221-30206	Acuity AccuRange 4000	OSI Laser Scan Auto Sense 800	ASIM TT 293
General	Height	4–20 m	0–16.5 m	6–10 m	5–6 m
	Power supply	24 V (1.8 A)	12–48 V (2 A)	12–15 V (0.5 A)	12 V (0.1 A)
	Consumption	20 W	-	40 W	-
	Emitter type	IR Diode	IR Diode	InGaAs Diode	PIR
	Wavelength	880 nm	780 nm	904 nm	8-14 µm
	Maximum power	1 mW	8 mW	1 mW	-
	Interface type	RS232/RS422	RS232/RS485	RS232/RS422	RS485
Communications	Transfer rate	19.2 K/38.4 K	300–38.4 K	19.2–57.6 K	9.6 K
	High speed option	500 K (RS422)	50 K (RS422)	1.25 M (RS422)	-
	Angular resolution	0.25°/0.50°/1°	0.08°	0.65°	-
	Response time	53/26/13 ms	80 ms	2.7 ms	-
	Scanning frequency	75 Hz	12 Hz	360 Hz	-
Scanner	Scanning angle	180°	360°	55° (7 m)	36° (6 m)
	Resolution	10 mm	12.5 mm	5 mm	-
Others	Weight	9 kg	1.6 kg	13.6 kg	1.8 kg

**Table 2 sensors-21-02082-t002:** Measurement ranges.

Measurement Range (m)	Bits	Maximum Measured Value (m)
8	13	8183
16	14	16,385
32	15	32,759
80	16	8183

**Table 3 sensors-21-02082-t003:** Specifications achieved.

Features	
Height above the track (m)	8–10
Maximum consumption (W)	40
Maximum speed	128 km/h
Laser class	I (safe)
Interface type	RS232/RS495/USB
Baud rate, transfer rate (KBd)	9.6–500
Scanning Frequency (Hz)	15–100
Response time (ms)	53–10
Scanning angle (°)	100°–180°
Angular resolution (°)	0.25°–1°

**Table 4 sensors-21-02082-t004:** Predictive parameters.

Parameter	Meaning
Width	Width of each vehicle.
Height	Maximum Value obtained from the 3D profile.
Ratio XCF	Number of Values between the center of the vehicle and the end that present a Value greater than X% of the maximum of the profile. In this group, there are five parameters: Ratio 50CF, Ratio 60CF, Ratio 70CF, Ratio 80CF, and Ratio 90CF, representing respectively at 50%, 60%, 70%, 80%, and 90% of the maximum Value of the profile or height.
Ratio XIC	Number of Values between the beginning and the center of the vehicle that present a Value greater than X% of the profile maximum. In this group, there are five parameters: Ratio 50IC, Ratio 60IC, Ratio 70IC, Ratio 80IC, and Ratio 90IC.
Ratio 100–125	Number of Values between the start of the vehicle and its center that presents Values between 1000 and 1250 m.
Value 3XY	After dividing the vehicle profile into two parts (considering three parts: beginning, center, and end), this Value shows the number resulting from dividing the profile Value on X by the profile Value on Y. These are Value 321 and Value 332.
Value 5XY	After dividing the vehicle profile into four parts (considering five parts), this Value shows the number resulting from dividing the profile Value on X by the profile Value on Y. In this group, there are the parameters: Value 532 and Value 543.
Value 7XY	After dividing the profile of the vehicle into six parts (considering seven parts), this Value shows the number resulting from dividing the profile Value on X by the profile Value on Y. In this group, there are Value 724, Value 734, Value 754, and Value 764
Der I50	Calculation of the derivative of the profile between the initial and the central Value of the vehicle profile.
Der 50F	Calculation of the derivative of the profile between the central Value and the end of the vehicle profile.
Range X–Y	Number of profile data between X and Y (in centimeters). Thus, the parameter Range 0–100 represents the number of profile data between 0 and 1 m; Range 50–100 is the number of profile data between 0.50 and 1 m. In this group, there are Range 0–100, Range 50–100, Range 100–125, Range 100–150, Range 150–175, Range 150–200, Range 200–250, and Range 250–300.
Range 300	Number of profile data greater than 3 m.

## Data Availability

Data is contained within the article or Appendix A.

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
