# Peer review of "Design, Implementation, and Configuration of Laser Systems for Vehicle Detection and Classification in Real Time"

_sensors, 2021, doi:10.3390/s21062082_

Round 1
Reviewer 1 Report
This version of the paper is OK to this reviewer.
Author Response
Thank you very much for your words. After doing such hard work, we appreciate receiving this feedback.
Currently your opinion coincides with that of two other reviewers, but a fourth required us to make a series of changes. Therefore I guess you will be contacted again to review the article one more time. This is basically the same one you reviewed, but even more complete. Then, we do not believe that you have any problem in qualifying it as 'ready to publish'.
Thank you very much in advance for everything.
Reviewer 2 Report
I would like to thank the authors for the effort made to include all the modifications in the paper. However, from the point of view of a scientific paper with clear unpublished contributions, this work still has two major limitations:
1) Original scientific contributions are still unclear. The paper is more focused on a technological advance, presenting guidelines for some stages of the proposed process, but it lacks a more solid scientific approach, such as mentioned related works in the field.
2) Comparisons with other methods. The authors only mention other papers, but they do not include comparisons in the results. With that, it is harder to identify the relevance of proposed steps and the advantage of the method as a whole.
Based on those remarks, I believe that the article cannot be accepted in its present form.
Author Response
Thank you very much for the time invested in reviewing this paper. Your feedback is appreciated.
1) Original scientific contributions are still unclear. The paper is more focused on a technological advance, presenting guidelines for some stages of the proposed process, but it lacks a more solid scientific approach, such as mentioned related works in the field. --> We believe that any reader with technical knowledge can easily identify the scientific contribution of this research after reading the paper.
As the abstract indicates, this system does not try to present all the machine learning behind the system (algorithms, signal processing, etc ...). All this information will be presented in a new paper, since it involves a great statistical analysis and a complex mathematical model. In this way, this paper simply tries to present a laser system optimized for road environments, which, as it is implemented from scratch, can follow as a guide for new models. This is how it is detailed in the abstract and in the introduction. Therefore we do not believe that there can be room for confusion.
It is true that there is a lot of essential information that has not been included in this paper, but it should be noted that this research lasted more than 3 years and gave rise to two doctoral theses. Therefore, we have first decided to present the system in order to later delve into the rest of the technicalities.
It was a three years research in which the Polytechnic University of Valencia and the Valencia City Council participated. In fact, the system continues functioning in the city and providing data of great interest. Therefore, we consider that there is no greater proof of its reliability and robustness than this fact.
Leaving aside the techniques of signal processing, it could be said that the main contribution of this system (this paper) would be:
- The optimization of a technology (widely extended but little used in the ITS sector) to be used in the road sector.
-The addition of functionalities to a sensor that nowadays is basically used only to measure the speed of vehicles and punish them if necessary.
-The homogenization of a technology that currently requires specific software and hardware depending on the laser used, which hinders its expansion.
-The design of hardware and software that allows vehicles to be monitored independently of the laser used (the possibilities that this offers are innumerable).
-The implementation of a laser system that works with 3D magnetic profiles, performs very powerful signal processing and has proven to work flawlessly under different conditions in the city of Valencia.
However, in order to clarify all this information, we have rewritten certain parts of the manuscript and added others (can be seen in 'Manuscript with track changes'). We hope that this time they meet your expectations.
2) Comparisons with other methods. The authors only mention other papers, but they do not include comparisons in the results. With that, it is harder to identify the relevance of proposed steps and the advantage of the method as a whole. --> Our research group, which has more than twenty years of experience in ITS, is a specialist in magnetic loops. In fact, we have currently obtained a patent related to loops (with a particular geometry) to be installed in bike lanes to monitor personal mobility vehicles. These are some of our latest papers about loops:
[1] Mocholí-Belenguer, Ferran; Martinez-Millana, Antonio; Mocholí Salcedo, Antonio; Arroyo Núñez, José Humberto. (2020) Vehicle Identification by Means of Radio-Frequency-Identification Cards and Magnetic Loops. IEEE Transactions on Intelligent Transportation Systems, 12 (21), 5051 - 5059. 10.1109/TITS.2019.2948221
[2] Mocholí-Belenguer, Ferran; Mocholí Salcedo, Antonio; Guill Ibáñez, Antonio; Milian Sanchez, Victor . (2019) Advantages offered by the double magnetic loops versus the conventional single ones. PLoS ONE, 2 (14), 1 - 24. 10.1371/journal.pone.0211626
[3] Mocholí-Belenguer, Ferran; Martinez-Millana, Antonio; Mocholí Salcedo, Antonio; Milián Sánchez, Victor. (2019) Vehicle modeling for the analysis of the response of detectors based on inductive loops. PLoS ONE, 9 (14), 1 - 28. 10.1371/journal.pone.0218631
[4] Mocholí-Belenguer, Ferran; Mocholí Salcedo, Antonio; Milian Sánchez, Victor; Arroyo Nuñez, José Humberto. (2018) Double Magnetic Loop and Methods for Calculating Its Inductance. Journal of Advanced Transportation (218)1 - 15. 10.1155/2018/6517137
[5] Mocholí Salcedo, Antonio; Arroyo-Núñez, José Humberto; Milian-Sanchez, Victor M.; Palomo-Anaya, M Jose; Arroyo-Nunez, Alexander. (2017) Magnetic Field Generated by the Loops Used in Traffic Control Systems. IEEE Transactions on Intelligent Transportation Systems, 8 (18), 2126 - 2136. 10.1109/TITS.2016.2632972
[6] Mocholí Salcedo, Antonio; Arroyo Núñez, José Humberto; Victor Milian Sanchez; Verdú Martín, Gumersindo Jesús; Arroyo Nuñez, Alexander. (2017) Traffic Control Magnetic Loops Electric Characteristics Variation Due to the Passage of Vehicles Over Them. IEEE Transactions on Intelligent Transportation Systems, 6 (18), 1540 - 1548.
As you can see, some of them are even published in a first decile journals (second in the transport category and with an impact factor of almost 7). Therefore, it is evident that we could write a whole paper indicating the advantages and disadvantages of each sensor and discuss when it is better to use one and the other.
However, we consider that in a paper in which it is intended to present the system, specify its benefits and demonstrate its robustness, a great discussion on different systems would be out of place. In addition, it should be taken into account that today there are countless ITS systems. Then, what sensor do we compare it to? With magnetic loops? With microwave radars? With magnetometers? ... This was the reason why we decided to compare the performance of different commercial laser models. We thought it was the best option.
However, in this new version we have modified many aspects so that it finally meets your expectations and you can contribute to the publication of this.
Thank you very much for your attention and work.
Best regards,
Ferran Mocholí Belenguer.
Reviewer 3 Report
Overall the paper is difficult to read and can be improved in all aspects.The paper describes how a specific LIDAR system was deployed on one street in Valencia, Spain, for vehicle classification and counting.
The paper suffers from few major flows: 1) it does not clearly present the scope of the research 2) the methods lack scientific rigor and 3) the results lack the necessary details.
On the form the outline can be improved to help clarify, and improving the english can help the reader. To give a few examples that use improper words, line 42 "challenges are ancient", line 46 "crowdedness creates discomfort for users". More importantly, there are many unsubstantiated, generic and disconnected statements, e.g., at the end of the introduction in the paragraphs starting at line 104 one would expect to see a clear explanation on the scope of the research and contributions. Figures and tables are difficult to read and not well formatted, lists and figures seem taken from PPT presentations. The paper reads more as an internal report with many unnecessary details and unjustified choices, than a research paper. Some acronyms such as PCB are not introduced which is problematic for readers that are not familiar with the domain.
Prior and related work should be more thorough, e.g., there are several very recent papers on the same topic and related (double induction loop detectors used in truck classification) that one would expect to see cited and compared with.
The focus is unclear with generic comments, too much details on implementation specific items and not enough on overall system design, requirements, and validation metrics.
One of the main reasons to use LIDAR seems to be to discriminate vehicle class especially for freight as freight impacts the infrastructure, traffic and health (source of pollution). Adding more consideration to explain how the sensor is set-up and the adequacy of the deployment in relation (if sensing heavy articulated trucks that are most impactful and originating on port / warehouse areas is important then it is questionable weather the deployment as tested is useful).
There is no comparison with other sensors/baseline (how LIDAR is an improvement with respect to a double loop detector for example?). The classifications results presented are not sufficiently explained to allow reproducing (e.g., how the ground truth was determined, is the dataset available, what is its size...) Other question could be addressed: accuracy of speed measurements, truck class are often based on number of axels, can LIDAR detect those classes? More importantly, the references given for the classifier are from the authors, one a PHD thesis from 2011 and the other from a publication without date and publisher. Also both citations are in Spanish which complicates accessibility for non-Spanish speakers.
There are points mentioned that are not sufficiently developed to be useful, e.g., deployment costs, traffic analytics, IoT (e.g., there is a clear contribution in making the data from the test unit available remotely in real-time but no discussion on scalability, compatibility with existing systems, and IoT features such as processing closer to the sensor).
Author Response
Thank you very much for the time invested in reviewing this paper. Your feedback is appreciated. We respond to your comments below:
The paper suffers from few major flows: 1) it does not clearly present the scope of the research 2) the methods lack scientific rigor and 3) the results lack the necessary details. --> We believe that any reader with technical knowledge can easily identify the scientific contribution of this research after reading the paper.
As the abstract indicates, this system does not try to present all the machine learning behind the system (algorithms, signal processing, etc ...). All this information will be presented in a new paper, since it involves a great statistical analysis and a complex mathematical model. In this way, this paper simply tries to present a laser system optimized for road environments, which, as it is implemented from scratch, can follow as a guide for new models. This is how it is detailed in the abstract and in the introduction. Therefore we do not believe that there can be room for confusion.
It is true that there is a lot of essential information that has not been included in this paper, but it should be noted that this research lasted more than 3 years and gave rise to two doctoral theses. Therefore, we have first decided to present the system in order to later delve into the rest of the technicalities in other papers.
It was a three years research in which the Polytechnic University of Valencia and the Valencia City Council participated. In fact, the system continues functioning in the city and providing data of great interest. Therefore, we consider that there is no greater proof of its reliability and robustness than this fact.
Leaving aside the techniques of signal processing, it could be said that the main contribution of this system (this paper) would be:
- The optimization of a technology (widely extended but little used in the ITS sector) to be used in the road sector.
-The addition of functionalities to a sensor that nowadays is basically used only to measure the speed of vehicles and punish them if necessary.
-The homogenization of a technology that currently requires specific software and hardware depending on the laser used, which hinders its expansion.
-The design of hardware and software that allows vehicles to be monitored independently of the laser used (the possibilities that this offers are innumerable).
-The implementation of a laser system that works with 3D magnetic profiles, performs very powerful signal processing and has proven to work flawlessly under different conditions in the city of Valencia.
Our research group, which has more than twenty years of experience in ITS, is a specialist in magnetic loops. In fact, we have currently obtained a patent related to loops (with a particular geometry) to be installed in bike lanes to monitor personal mobility vehicles. These are some of our latest papers about loops:
[1] Mocholí-Belenguer, Ferran; Martinez-Millana, Antonio; Mocholí Salcedo, Antonio; Arroyo Núñez, José Humberto. (2020) Vehicle Identification by Means of Radio-Frequency-Identification Cards and Magnetic Loops. IEEE Transactions on Intelligent Transportation Systems, 12 (21), 5051 - 5059. 10.1109/TITS.2019.2948221
[2] Mocholí-Belenguer, Ferran; Mocholí Salcedo, Antonio; Guill Ibáñez, Antonio; Milian Sanchez, Victor . (2019) Advantages offered by the double magnetic loops versus the conventional single ones. PLoS ONE, 2 (14), 1 - 24. 10.1371/journal.pone.0211626
[3] Mocholí-Belenguer, Ferran; Martinez-Millana, Antonio; Mocholí Salcedo, Antonio; Milián Sánchez, Victor. (2019) Vehicle modeling for the analysis of the response of detectors based on inductive loops. PLoS ONE, 9 (14), 1 - 28. 10.1371/journal.pone.0218631
[4] Mocholí-Belenguer, Ferran; Mocholí Salcedo, Antonio; Milian Sánchez, Victor; Arroyo Nuñez, José Humberto. (2018) Double Magnetic Loop and Methods for Calculating Its Inductance. Journal of Advanced Transportation (218)1 - 15. 10.1155/2018/6517137
[5] Mocholí Salcedo, Antonio; Arroyo-Núñez, José Humberto; Milian-Sanchez, Victor M.; Palomo-Anaya, M Jose; Arroyo-Nunez, Alexander. (2017) Magnetic Field Generated by the Loops Used in Traffic Control Systems. IEEE Transactions on Intelligent Transportation Systems, 8 (18), 2126 - 2136. 10.1109/TITS.2016.2632972
[6] Mocholí Salcedo, Antonio; Arroyo Núñez, José Humberto; Victor Milian Sanchez; Verdú Martín, Gumersindo Jesús; Arroyo Nuñez, Alexander. (2017) Traffic Control Magnetic Loops Electric Characteristics Variation Due to the Passage of Vehicles Over Them. IEEE Transactions on Intelligent Transportation Systems, 6 (18), 1540 - 1548.
As you can see, some of them are even published in a first decile journals (second in the transport category and with an impact factor of almost 7). Therefore, it is evident that we could write a whole paper indicating the advantages and disadvantages of each sensor and discuss when it is better to use one and the other.
However, we consider that in a paper in which it is intended to present the system, specify its benefits and demonstrate its robustness, a great discussion on different systems would be out of place. In addition, it should be taken into account that today there are countless ITS systems. Then, what sensor do we compare it to? With magnetic loops? With microwave radars? With magnetometers? ... This was the reason why we decided to compare the performance of different commercial laser models. We thought it was the best option.
However, in this new version we have modified many aspects (can be seen in 'Manuscript with track changes') so that it finally meets your expectations and you can contribute to the publication of this.
Thank you very much for your attention and work.
On the form the outline can be improved to help clarify, and improving the english can help the reader. To give a few examples that use improper words, line 42 "challenges are ancient", line 46 "crowdedness creates discomfort for users". --> Thank you for the suggestion. As English is not our mother tongue, we welcome grammar suggestions.
However, these introductory paragraphs have been rewritten and, this particular part, removed. In addition, a complete check of the writing and grammar of the rest of the paper has also been carried out.
More importantly, there are many unsubstantiated, generic and disconnected statements, e.g., at the end of the introduction in the paragraphs starting at line 104 one would expect to see a clear explanation on the scope of the research and contributions. --> The line 104 that you comment is precisely in the 'Introduction' section. This is where we introduce the physical principle of laser technology and explain how they work and their applications. Therefore, we consider that there is no place at this point to enumerate the contributions of our systems. These are detailed in 'materials and methods' sections and, specially, in 'results' section.
Figures and tables are difficult to read and not well formatted, lists and figures seem taken from PPT presentations. --> Thank you for the annotation.
It is true that some figures have been extracted from PDF documents (such as the two doctoral theses with which this research culminated). We really consider that these are easy to read, like the tables, however, we have made significant changes so that these now do meet your expectations.
Some acronyms such as PCB are not introduced which is problematic for readers that are not familiar with the domain. --> Thank you for the detail. You are right. As they are widely known concepts in the scientific community, we forgot to introduce them properly. However, this term as well as many others (PIR, LIDAR, etc ...) have been introduced in detail for readers that are not familiar with ITS systems.
Prior and related work should be more thorough, e.g., there are several very recent papers on the same topic and related (double induction loop detectors used in truck classification) that one would expect to see cited and compared with. -->
Our research group, which has more than twenty years of experience in ITS, is precisely a specialist in magnetic loops. These are some of our latest papers about loops:
[1] Mocholí-Belenguer, Ferran; Martinez-Millana, Antonio; Mocholí Salcedo, Antonio; Arroyo Núñez, José Humberto. (2020) Vehicle Identification by Means of Radio-Frequency-Identification Cards and Magnetic Loops. IEEE Transactions on Intelligent Transportation Systems, 12 (21), 5051 - 5059. 10.1109/TITS.2019.2948221
[2] Mocholí-Belenguer, Ferran; Mocholí Salcedo, Antonio; Guill Ibáñez, Antonio; Milian Sanchez, Victor . (2019) Advantages offered by the double magnetic loops versus the conventional single ones. PLoS ONE, 2 (14), 1 - 24. 10.1371/journal.pone.0211626
[3] Mocholí-Belenguer, Ferran; Martinez-Millana, Antonio; Mocholí Salcedo, Antonio; Milián Sánchez, Victor. (2019) Vehicle modeling for the analysis of the response of detectors based on inductive loops. PLoS ONE, 9 (14), 1 - 28. 10.1371/journal.pone.0218631
[4] Mocholí-Belenguer, Ferran; Mocholí Salcedo, Antonio; Milian Sánchez, Victor; Arroyo Nuñez, José Humberto. (2018) Double Magnetic Loop and Methods for Calculating Its Inductance. Journal of Advanced Transportation (218)1 - 15. 10.1155/2018/6517137
[5] Mocholí Salcedo, Antonio; Arroyo-Núñez, José Humberto; Milian-Sanchez, Victor M.; Palomo-Anaya, M Jose; Arroyo-Nunez, Alexander. (2017) Magnetic Field Generated by the Loops Used in Traffic Control Systems. IEEE Transactions on Intelligent Transportation Systems, 8 (18), 2126 - 2136. 10.1109/TITS.2016.2632972
[6] Mocholí Salcedo, Antonio; Arroyo Núñez, José Humberto; Victor Milian Sanchez; Verdú Martín, Gumersindo Jesús; Arroyo Nuñez, Alexander. (2017) Traffic Control Magnetic Loops Electric Characteristics Variation Due to the Passage of Vehicles Over Them. IEEE Transactions on Intelligent Transportation Systems, 6 (18), 1540 - 1548.
As you can see, some of them are even published in a first decile journals (second in the transport category and with an impact factor of almost 7). In addition, as researchers in the field we are aware of the new and most recognized scientific paper on loops and ITS systems. Therefore, it is evident that we could write a whole paper indicating the advantages and disadvantages of each sensor and discuss when it is better to use one and the other.
However, we consider that in a paper in which it is intended to present the system, specify its benefits and demonstrate its robustness, a great discussion on different systems would be out of place. In addition, it should be taken into account that today there are countless ITS systems. Then, what sensor do we compare it to? With magnetic loops? With microwave radars? With magnetometers? ... This was the reason why we decided to compare the performance of different commercial laser models. We thought it was the best option.
However, in this new version we have modified many aspects (can be seen in 'Manuscript with track changes') so that it finally meets your expectations and you can contribute to the publication of this.
Thank you very much for your attention and work.
One of the main reasons to use LIDAR seems to be to discriminate vehicle class especially for freight as freight impacts the infrastructure, traffic and health (source of pollution). Adding more consideration to explain how the sensor is set-up and the adequacy of the deployment in relation (if sensing heavy articulated trucks that are most impactful and originating on port / warehouse areas is important then it is questionable weather the deployment as tested is useful). --> This idea is very interesting. One of the functionalities of this system is precisely the ability to detect and classify vehicles in real time. Therefore, it could be applied to what you comment.
However, as indicated in the paper, this system was installed in three locations in the city. All of them very close to the city center. Therefore, unfortunately the traffic of these vehicles could not be analyzed with the aim that you propose, since they usually circulate on interurban roads. But, of course this system could help in the detection of heavy articulated trucks that are more impactful and originate in port / depot areas.
We take it as an annotation to carry out a research on the matter and publish the results perhaps in a congress. Thanks for the idea.
There is no comparison with other sensors/baseline (how LIDAR is an improvement with respect to a double loop detector for example?). --> This question has been answered in a previous comment.
Other question could be addressed: truck class are often based on number of axels, can LIDAR detect those classes? --> As indicated in the paper, this system allows to classify vehicles according to a German standar, which discriminates 8 + 1 vehicle (8 categories plus an extra category called 'unknown'). At no time is it mentioned that the system is capable of detecting truck axles. Therefore, we believe that it is obvious that if it is not mentioned, the system can not do it.
However, we would like to indicate that, by modifying the microcontroller programming and adding new algorithms and signal processing, this axis detection would be possible. This functionality has simply not been implemented because it was not the goal of the system.
There are points mentioned that are not sufficiently developed to be useful, e.g., deployment costs, traffic analytics, IoT (e.g., there is a clear contribution in making the data from the test unit available remotely in real-time but no discussion on scalability or compatibility with existing systems --> After figure 13, it is indicated that "the system was designed so that these results can be sent to traffic management centers in real time using any type of communication technology: fiber optic cable, coaxial cable, GPRS, GSM, WiFi or WiMAX. In fact, during field tests in Malilla, WiMAX technology was used to communicate with the Valencia City Council traffic control room from the system site." And in sections 4 and 5 (discussion and conclusions) information is added in this regard.
Therefore, we consider that the requested information is already available in the paper. However, we have made general changes to the manuscript so that it meets your expectations.
Thank you very much for the time invested in reviewing the article. Your feedback has helped us to develop a version of higher quality and interest for Sensors readers. We hope that you value all the work done and contribute to this research to be finally published.
Reviewer 4 Report
Contribution:
Design, implementation and configuration of laser systems to obtain 3D profiles of vehicles, which collect more precise information about the state of the roads. Also, a series of criteria to help make these crucial and difficult decisions.
Major issues:
First, the authors provide an excellent and detailed discussion of a solid design, development, and field test of an important ITS system. In the context of the information they have provided, the authors have done a good job. That said, I have one significant issue with the paper which prevents me from recommending publication. The heart of the issue, based upon the author statements as well as knowledge of the field, is the processing of the LIDAR data. The authors mention use of Machine Learning (ML). However, while there is detailed discussion of the hardware design (PCB, etc.), there is essentially no discussion of their use of ML. I believe the authors need to consider including some or all of the following in much more detail:
* The specif ML approach used
* Any training for the ML, including info on training data set(s)
* How did they implement the ML
* More info on specifically why they use ML
* What would be the situation without ML? Non-real time? No capability at all? Slower?
* How does the authors' use of ML improve on prior use, or is ML a tool here without theoretical improvement (which is fine)
I may have missed the point of the authors' work, but if not, then I could not recommend this paper without most if not all of this info.
For Fig. 1, please explain "lap time". It isn't clear just from the figure. I think this is important to your system, so worth elaborating.
As machine learning is central to your approach, consider adding this to the title.
line 142, "It must discriminate up to 8 + 1 vehicle categories". Is there some reason you don't just say "9"? I'm not familiar with the German standard. Does the '+1" relate to trailers?'
line 177, who is the reviewer in Table 1 (last line)? If appropriate, include a reference.
line 199, "eliminate any possible dependence on the sensor equipment". Is it fair to say "eliminate", or better to say "minimize"? Your call, you know your design.
line 204, this may be a good paragraph to describe "lap time"
line 242, "it is necessary to consider both possible irregularities in the asphalt and static objects located on the road" Here, also sensor noise? Or is this small enough to neglect?
Minor issues:
Throughout, always follow the same pattern to define acronyms on first use. E.g. on line 265, use "printed circuit board (PCB)". I.e., actively define the acronym right at the start, do not just assume it later.
line 91, "Fig. 2 shows". Check journal guidance, is this OK at start of sentence?
line 99, "(b)." -> "(b)"
line 120 and throughout. Be consistent. Use "LIDAR", not "lidar". It's an acronym, and always uppercase.
line 139, "unaffected as possible to inclement" -> "unaffected as possible by inclement"
line 170, "a PIR sensor". Define this acronym
Author Response
Thank you very much for your words. After conducting such a long and arduous research, your feedback is appreciated.
To facilitate communication, we comment on each of your comments individually.
Major issues:
First, the authors provide an excellent and detailed discussion of a solid design, development, and field test of an important ITS system. In the context of the information they have provided, the authors have done a good job. That said, I have one significant issue with the paper which prevents me from recommending publication. The heart of the issue, based upon the author statements as well as knowledge of the field, is the processing of the LIDAR data. The authors mention use of Machine Learning (ML). However, while there is detailed discussion of the hardware design (PCB, etc.), there is essentially no discussion of their use of ML. I believe the authors need to consider including some or all of the following in much more detail:
* The specif ML approach used
* Any training for the ML, including info on training data set(s)
* How did they implement the ML
* More info on specifically why they use ML
* What would be the situation without ML? Non-real time? No capability at all? Slower?
* How does the authors' use of ML improve on prior use, or is ML a tool here without theoretical improvement (which is fine)
I may have missed the point of the authors' work, but if not, then I could not recommend this paper without most if not all of this info. --> Thank you very much for your comment and for the time spent reviewing our article. We have worked really hard to carry out this study and therefore, it is a pleasure to hear
comments like yours.
You are right. It is true that the the paper could be even more complete and include a lot of information about machine learning, which is which is fundamental in the developed system. It can be seen that the algorithms and data processing performed by the system presented are not detailed in this paper. In fact, no equation or mathematical reference appears in this regard,
but there is a clear reason: all the signal processing developed entirely by our research group as well as the algorithms in charge of vehicle detection and classification are going to be explained in an upcoming paper. This is indeed already finished and almost ready to be sent for review.
Please keep in mind that these algorithms, statistical analyses and mathematical processes are very extensive and complex. Therefore, they could not be easily and briefly detailed in this paper. These are interesting and convoluted enough to be explained in detail in a new paper. In addition, we also consider that the paper itself is already quite extensive. Therefore, the detailed explanation of everything you comment would have no place in this paper, which, as
the abstract indicates, simply aims to present the system. Nevertheless, we added some additional information on this topic (including a table with the statistical parameters used in the vehicle detection and classification algorithm) for a better understanding.
We trust our explanation will meet your expectations.
For Fig. 1, please explain "lap time". It isn't clear just from the figure. I think this is important to your system, so worth elaborating. --> This suggestion has been raised and answered in a comment below.
As machine learning is central to your approach, consider adding this to the title. --> As we mentioned in the first comment, all the signal processing, algorithms and machine lerning will be presented in a new paper, which I can tell you it will be called "Machine learning techniques and advanced algorithms for the identification and classification of vehicles using laser sensors". Therefore, we consider that the title should not in principle include those words.
Thanks anyway for the suggestion.
line 142, "It must discriminate up to 8 + 1 vehicle categories". Is there some reason you don't just say "9"? I'm not familiar with the German standard. Does the '+1" relate to trailers?' --> We refer to this standard as '8+1' and not as '9' for various reasons. The first one is that this is how it is officially called, therefore, we simply refer to it as it is. The second is that this standard discriminates up to 8 different categories of vehicles (as shown in Figure 13 - motorcycle, car, car with trailer, van, truck, truck with trailer, bus and articulated), being this '+1' an extra category called 'unknown'. In this way, it is capable of discriminating 8 types of vehicles or classifying it as 'unknown'. Hence its official name of '8+1'.
However, thank you for your comments, because we have realized that this clarification had not been included in the paper. The information could be consulted in the included reference but we had not explained this detail. Therefore, in the new version it has been included.
Thank you for your comment.
line 177, who is the reviewer in Table 1 (last line)? If appropriate, include a reference. -->Table 1 shows the most important parameters regarding the commercial equipment studied in this paper. These features were obtained directly from their corresponding datasheet, as indicated in lines 171-172 (of the old version).
Therefore, the veracity of these data can be easily checked by consulting the datasheets of these equipments. However, we have included references so that this information can be quickly verified.
Thank you for your attention to detail and your great work
line 199, "eliminate any possible dependence on the sensor equipment". Is it fair to say "eliminate", or better to say "minimize"? Your call, you know your design. --> In this regard, we think you are absolutely right and endorse your view. These grammatical errors are due to the fact that English is not our mother tongue. However, thank you very much for the clarification. In the new version you will find 'minimize' instead of 'eliminate'.
Thank you for your contribution.
line 204, this may be a good paragraph to describe "lap time" --> Lines 78-79 of the old version very briefly described how laser technology works, as we assume that readers of this journal are knowledgeable in the field. It is said that 'The laser emits multiple pulses of infrared light, which bounce off the vehicle and return to it', which we thought that together with Figure 1 did not give rise to confusion. However, it is true that this concept is not explained during the paper. For this reason, in this new version we have included a short paragraph referring to it, just where you indicated.
line 242, "it is necessary to consider both possible irregularities in the asphalt and static objects located on the road" Here, also sensor noise? Or is this small enough to neglect? --> These systems are very powerful commercial equipment, therefore, they are fully optimized in terms of noise level. In fact, in Table 3 it can be seen that the complete system (with all the hardware added by us, that is, with greater 'noise sources') still offers very high performance. In addition, taking into account the size of the objects to be registered, the sensor noise, as you pointed out, is small enough to despise.
Thank you very much for your doubt.
Minor issues:
Throughout, always follow the same pattern to define acronyms on first use. E.g. on line 265, use "printed circuit board (PCB)". I.e., actively define the acronym right at the start, do not just assume it later. --> Thank you very much for detail. You are right. We made this mistake because we considered that it is a widely extended concept. However, in the new version this error has been corrected.
line 91, "Fig. 2 shows". Check journal guidance, is this OK at start of sentence? --> I have been checking journal guidance and I have not found anything indicating that it is not allowed at start of sentence. Therefore we consider that it is correct and accepted.
line 99, "(b)." -> "(b)" --> Thanks for the gesture. It has been corrected.
line 120 and throughout. Be consistent. Use "LIDAR", not "lidar". It's an acronym, and always uppercase. --> You are right. We forgot to write it that time in capital letters. Thank you for detail. It has been corrected in the new version.
line 139, "unaffected as possible to inclement" -> "unaffected as possible by inclement" --> Thanks for these grammar corrections. They are appreciated. In the new version of the manuscript this sentence has been rewritten.
line 170, "a PIR sensor". Define this acronym --> Thanks for the detail. As with other terms, we thought that this was widely known. However, the definition of this acronym has been added.
We wholeheartedly hope that our responses help you to appreciate our work a little more and to understand the hard work behind it. It was a 3 years research. Therefore, we would really appreciate if you consider our article ready for publication.
Best regards,
Ferran Mocholí Belenguer.
Round 2
Reviewer 2 Report
This reviewer thanks the authors for their responses, although the arguments pointed out are insufficient to change the paper's final evaluation.
The authors state that the paper presents two main contributions. The first one is related to the proposition of a guideline to obtain reliable information on vehicle traffic, detailing the following steps: (i) choose the most suitable type of laser, (ii) select its configuration, (iii) determine the optimal location, and (iv) process the information. Regarding this first contribution, this reviewer believes that, based on the reported results, it indicates a technological advance, showing how the infrastructure has been designed and does not discuss a relevant scientific proposition. The results section is more similar to a technical report than a scientific paper in a high impact factor journal.
The second contribution is the presentation of a complete laser system for vehicle detection and classification. When combined with the first contribution, it could represent relevant and original content, desirable for this journal. However, once again based on the reported results, it is highly difficult to observe the relevance pointed out by the authors due to the following factors:
1) The reported detection rate of almost 98% is vaguely compared with magnetic loops and microwave radars (only included in this last round of review). However, suitable state-of-the-art methods could be used as a reference to substantially indicate that this is a relevant result - some of them were listed by the authors in the references. Without that, it raises the doubt whether the whole apparatus built was essential to achieve such accuracy or some other state-of-the-art method already solves the proposed problem.
2) Most of the details regarding the recognition system are omitted and reserved for a future paper. This also makes it difficult to understand the proposal and the relevance of the presented results.
Finally, this reviewer does not question the authors' competence and the relevance of the topic. Based on a complete reading of the article, the notes raised here serve as suggestions for a more precise and reliable presentation of the results undoubtedly present in the research that the authors have conducted.
Author Response
The authors state that the paper presents two main contributions. The first one is related to the proposition of a guideline to obtain reliable information on vehicle traffic, detailing the following steps: (i) choose the most suitable type of laser, (ii) select its configuration, (iii) determine the optimal location, and (iv) process the information. Regarding this first contribution, this reviewer believes that, based on the reported results, it indicates a technological advance, showing how the infrastructure has been designed and does not discuss a relevant scientific proposition. The results section is more similar to a technical report than a scientific paper in a high impact factor journal.
The second contribution is the presentation of a complete laser system for vehicle detection and classification. When combined with the first contribution, it could represent relevant and original content, desirable for this journal. However, once again based on the reported results, it is highly difficult to observe the relevance pointed out by the authors due to the following factors:
1) The reported detection rate of almost 98% is vaguely compared with magnetic loops and microwave radars (only included in this last round of review). However, suitable state-of-the-art methods could be used as a reference to substantially indicate that this is a relevant result - some of them were listed by the authors in the references. Without that, it raises the doubt whether the whole apparatus built was essential to achieve such accuracy or some other state-of-the-art method already solves the proposed problem.
2) Most of the details regarding the recognition system are omitted and reserved for a future paper. This also makes it difficult to understand the proposal and the relevance of the presented results.
Thank you very much for the time spent on this second round of review. We honestly want to thank you for your comments. We agree with some of your suggestions. They have been very decisive in preparing a better version of the paper.
However, we would like to clarify the following:
1) It is clear that the work presented in this paper is innovative and relevant in the bibliography. In fact, two of the reviewers have told us that they have done several searches in Google Schoolar on systems similar to ours and have not found anything about it.
2) The comparison that you ask us regarding the margins of error has no place as you propose. We consider that, of course it is very important to compare the performance of the system implemented with known technology (such as magnetic loops or CCTV systems, which have been included in this review as requested). However, we consider that mentioning the features ​​and correctly reference where such information can be consulted is sufficient.
We are practically experts in magnetic loops. You can check all our latest publications about them here:
Mocholí-Belenguer, Ferran; Martinez-Millana, Antonio; Mocholí Salcedo, Antonio; Arroyo Núñez, José Humberto. (2020) Vehicle Identification by Means of Radio-Frequency-Identification Cards and Magnetic Loops. IEEE Transactions on Intelligent Transportation Systems, 12 (21), 5051 - 5059. 10.1109/TITS.2019.2948221
Mocholí-Belenguer, Ferran; Mocholí Salcedo, Antonio; Guill Ibáñez, Antonio; Milian Sanchez, Victor . (2019) Advantages offered by the double magnetic loops versus the conventional single ones. PLoS ONE, 2 (14), 1 - 24. 10.1371/journal.pone.0211626
Mocholí-Belenguer, Ferran; Martinez-Millana, Antonio; Mocholí Salcedo, Antonio; Milián Sánchez, Victor. (2019) Vehicle modeling for the analysis of the response of detectors based on inductive loops. PLoS ONE, 9 (14), 1 - 28. 10.1371/journal.pone.0218631
Mocholí-Belenguer, Ferran; Mocholí Salcedo, Antonio; Milian Sánchez, Victor; Arroyo Nuñez, José Humberto. (2018) Double Magnetic Loop and Methods for Calculating Its Inductance. Journal of Advanced Transportation (218)1 - 15. 10.1155/2018/6517137
Mocholí Salcedo, Antonio; Arroyo-Núñez, José Humberto; Milian-Sanchez, Victor M.; Palomo-Anaya, M Jose; Arroyo-Nunez, Alexander. (2017) Magnetic Field Generated by the Loops Used in Traffic Control Systems. IEEE Transactions on Intelligent Transportation Systems, 8 (18), 2126 - 2136. 10.1109/TITS.2016.2632972
Mocholí Salcedo, Antonio; Arroyo Núñez, José Humberto; Victor Milian Sanchez; Verdú Martín, Gumersindo Jesús; Arroyo Nuñez, Alexander. (2017) Traffic Control Magnetic Loops Electric Characteristics Variation Due to the Passage of Vehicles Over Them. IEEE Transactions on Intelligent Transportation Systems, 6 (18), 1540 - 1548. 10.1109/TITS.2016.2612579
In fact, we have just patented a system based on magnetic loops for monitoring personal mobility vehicles on bike lanes. Therefore, we know the technology very well.
However, we do not believe that it is possible to explain in this paper the methods of obtaining the signal from magnetic loops or even CCTV systems in detail, since this information would go beyond the scope of this paper and can be consulted by anyone interested in the references.
3) It is true that most of the details regarding the recognition system are omitted and reserved for a future paper, as you say. But we hope you understand that all this information has no place in this paper, which is only intended to present the system. However, at your request, we have included a lot of previous information (such as Table 4) about it. We consider that it would not make sense to publish the algorithms and the signal processing developed without having previously presented the system in its entirety.
4) In this second review, one reviewer suggested acceptance of the manuscript and another one requested minor changes. Therefore, in this third round, you may not see very significant changes. However, we would appreciate if you would give us a chance and appreciate the hard work done in this research of more than three years.
We hope that these replies and changes will satisfy your expectations and provide a higher quality to the manuscript.
Thank you very much again for your contribution. It has been a pleasure to have been in contact with you.
Reviewer 3 Report
I'd like to thank the authors for the updates made to the manuscript and for providing feedback to the reviews.
The manuscript is now improved as a result, however I have noted below some lingering concerns that can be addressed to further improve the manuscript.
First off thank you for addressing the english language.
The related work discussion was improved with additional description of inductance loop detectors in the introduction. Consider adding some additional references, for example I am aware of the TAMS system from ITS at UCI https://escholarship.org/content/qt12w5z2c8/qt12w5z2c8.pdf that uses double loops for precise truck classification). You may also want to consider citing CCTV cameras as another source of precise vehicle class, speed etc information but also with drawbacks.
Also consider expanding the prior work on laser scanners, there is a large body of work on mobile applications (autonomous vehicles) and as fixed sensors. A quick look at google scholar seem to show that there is no previous work similar to yours: with lessons learned of a complete implementation and validation of an overhead lidar system, over a long term multi-site deployment that is connected to an actual traffic monitoring system. If that is the case consider highlighting these contributions.
One of the reasons cited for using lidar is the ability to obtain height information. It would be useful to explain how that information is useful.
The presentation can be much improved, for example icons and pictures of Figure 3 are too small and may be unclear to readers, and the PPT style bullet lists seem inappropriate in a scientific publication. While the Figure 3 and the bullet point list that follows provide valuable insights they are given without much justification and references. For the small pictures in the Figure 3 consider using text that users not familiar with what is depicted in the pictures can understand. For the bullet list consider using dots instead of the arrows and clarify each bullet with explanations; here are some questions that come to mind as I read through the list starting in line 167:
"It must be able to function properly regardless of its placement and laser sensor used." which contradicts the experimental set-up that uses sensors overhead. Placing the sensor overhead makes sense in several ways, and seems unique to this research.
"It must be intuitive and easy to use." why ease of use is important and in what sense, if the application calls for automated monitoring? I do see that there is a need for ease of access to the data for real-time monitoring.
"Automatic calibration must be available whenever possible." Why whenever possible? Why these sensors need calibration? This is the only mention to calibration in the paper, is that a concern and how was that addressed?
"It must be as unaffected as possible by inclement weather.", consider rephrasing and add context; e.g., is rain, snow a concern with these sensors? Can it be addressed with data processing?
"It must discriminate up to 8 + 1 vehicle categories according to the classification of the German TLS standard defined by the Federal Institute for Highway Research...". As I pointed out in the first review this is an example that makes the paper read like a report. Is this part of your requirements or did you pick this one because it is adopted by Valencia?
"It must be able to provide information on vehicle detection, classification, lane position, presence of tow bar, length, width, height, speed, opposite direction and traffic flow." again this seems a list of requirements for your specific implementation. I can see how some of these are used for traffic flow monitoring but things like tow bar and vehicle dimensions I am not sure. Sounds like these are features that traditional ML would use as features to determine vehicle class. Is that the reason?
The Figure 3 could serve as an outline for the paper. Consider adding some text to present the outline outline so readers can get situated.
In Section 2.4 System implementation consider clarifying the scope of the hardware early on: it just says "connect to a computer" - are there no existing commercial solutions? Is that for the experiments? Consider adding some introduction, it might not be obvious for the reader that it needed to process PC data and connect with a traffic monitoring system.
Thank you for improving the tables formatting. Consider using larger font, centering the text vertically and avoid smaller colored text, using multiple colors, especially low color contrast text (see Table 1).
I am not sure what the Fig 14 is about. Is that a monitoring dashboard that displays real-time information from the PCB? Consider clarifying.
You mention that direction is important to capture - are you monitoring both directions? For the results section consider providing more details on the dataset: e.g., hours recorded, number of PC profiles captured, weather conditions encountered, hours of ground truth video and any other information that relates to the actual system performance like downtown.
On the conclusions, consider clarifying the contributions (lessons learned, system hardware, validation over a year...) and suggest future work. Again the arguments made in conclusion can be clarified, e.g., "The addition of functionalities to a sensor that nowadays is basically used only to measure the speed of vehicles and fine them if necessary." is difficult to justify given the related work.
Author Response
I'd like to thank the authors for the updates made to the manuscript and for providing feedback to the reviews. The manuscript is now improved as a result, however I have noted below some lingering concerns that can be addressed to further improve the manuscript. First off thank you for addressing the english language. --> Thank you very much for the time spent on this second round of review. We honestly want to thank you for your comments and contributions. We fully agree with your suggestions. They have been very decisive in preparing a better version of the paper.
The related work discussion was improved with additional description of inductance loop detectors in the introduction. Consider adding some additional references, for example I am aware of the TAMS system from ITS at UCI https://escholarship.org/content/qt12w5z2c8/qt12w5z2c8.pdf that uses double loops for precise truck classification). You may also want to consider citing CCTV cameras as another source of precise vehicle class, speed etc information but also with drawbacks. --> In the last revision we tried to improve the discussion of the related work with a further description of inductance loop detectors. However, we decided not to include more technologies precisely because of the infinity of possibilities that exist. We were wondering: should we compare it with microwave radars, which are a worldwide standard? Should we compare it to machine vision cameras, which seems to be the road system of the future? But we dismiss it for the aforementioned reason: there are many road sensors currently on the market. Therefore, the discussion could become very extensive.
The TAMS system has been very interesting to us. We did not know. However, we consider that a discussion dedicated only to this system would have no place in our paper, since we believe that a generalized discussion about the different sensors is more interesting for readers than a specific system, which many may not know. In this sense, we have only added the reference in the corresponding paragraph so that readers can consult if they wish.
We have also added two large paragraphs including CCTV and the disadvantages they present compared to the developed laser system. We honestly believe that the added information is of great interest for readers, therefore we hope that this time it will meet your expectations.
Also consider expanding the prior work on laser scanners, there is a large body of work on mobile applications (autonomous vehicles) and as fixed sensors. A quick look at google scholar seem to show that there is no previous work similar to yours: with lessons learned of a complete implementation and validation of an overhead lidar system, over a long term multi-site deployment that is connected to an actual traffic monitoring system. If that is the case consider highlighting these contributions. --> Of course we have taken a lot of looks at google schoolar to verify that there are no similar systems. For this reason, we firmly believe in the result of the research and consider that this paper has sufficient potential to be published.
In the last review, we added a lot of information in conclusions section that corroborated the importance and innovation of our system. However, in this review we have still included more details.
Thank you very much for detail.
One of the reasons cited for using lidar is the ability to obtain height information. It would be useful to explain how that information is useful. --> This information has been included in the conclusions sections.
Thank you very much for detail.
The presentation can be much improved, for example icons and pictures of Figure 3 are too small and may be unclear to readers, and the PPT style bullet lists seem inappropriate in a scientific publication. While the Figure 3 and the bullet point list that follows provide valuable insights they are given without much justification and references. For the small pictures in the Figure 3 consider using text that users not familiar with what is depicted in the pictures can understand. For the bullet list consider using dots instead of the arrows and clarify each bullet with explanations; --> This figure has not been extracted from any PPT. In fact, it was done by one of our researchers. However, it is true that the figures are too small and perhaps difficult for readers to interpret. Therefore, in the new version we have included a new figure 3.
We hope you find this new figure better than the previous one.
Thank you for the suggestion.
here are some questions that come to mind as I read through the list starting in line 167:
"It must be able to function properly regardless of its placement and laser sensor used." which contradicts the experimental set-up that uses sensors overhead. Placing the sensor overhead makes sense in several ways, and seems unique to this research. --> We do not want this to seem exclusive to our research. With this phrase we want to indicate that one of the design premises of our system is that 'it must be able to function correctly regardless of the sensor used and where it is installed'. This is achieved thanks to the hardware and software implemented, which is actually exclusive to our research. It allows adaptation of the signals received from different sensors to a standard frame. Futhermore, it considers the different communication standards so that this can be done independently of the sensor.
Thank you for the doubt.
"It must be intuitive and easy to use." why ease of use is important and in what sense, if the application calls for automated monitoring? I do see that there is a need for ease of access to the data for real-time monitoring. --> We work daily developing products for different municipalities and even private companies, and we are aware that end customers want simple software that can be handled by anyone regardless of their knowledge.
Then, with this phrase we simply want to emphasize that, despite the internal complexity of the system, the graphical software application must be easy to use. In fact, Figure 14 of the manuscript shows this. It has a simple appearance where vehicles are counted, the 3D profiles are shown, etc... but in a very intuitive and visual way.
From our experience, it is often more difficult to display information as customers want than to obtain it.
"Automatic calibration must be available whenever possible." Why whenever possible? Why these sensors need calibration? This is the only mention to calibration in the paper, is that a concern and how was that addressed? --> Calibration of these systems should be carried out if the condition of the road has changed. That is, if, for example, there were a new obstacle in it. However, this situation does not usually occur as the authorities quickly pick up the obstacle and remove it from the road. In all other situations it is not necessary to calibrate more than the first time.
However, it is true that the phrase can lead to confusion, so we decided to rewrite it as: calibration must be available whenever required.
Thank you so much for detail.
"It must be as unaffected as possible by inclement weather.", consider rephrasing and add context; e.g., is rain, snow a concern with these sensors? Can it be addressed with data processing? --> You are right. This has been rewriten as you suggest.
Thank you.
"It must discriminate up to 8 + 1 vehicle categories according to the classification of the German TLS standard defined by the Federal Institute for Highway Research...". As I pointed out in the first review this is an example that makes the paper read like a report. Is this part of your requirements or did you pick this one because it is adopted by Valencia? --> This is a requirement that we set for our system. Currently, this German standard is the most widespread in Europe for classifying vehicles, as well as one of the most extensive (there are some others that only discriminate 5 categories of vehicles: motorcycles, cars, vans, buses and trucks).
Therefore, we consider that our system should be capable of, at least, providing the same data as the most powerful standard currently in Europe.
"It must be able to provide information on vehicle detection, classification, lane position, presence of tow bar, length, width, height, speed, opposite direction and traffic flow." again this seems a list of requirements for your specific implementation. I can see how some of these are used for traffic flow monitoring but things like tow bar and vehicle dimensions I am not sure. Sounds like these are features that traditional ML would use as features to determine vehicle class. Is that the reason? -->No, it is not. All these data, which are not so common in other ITS systems, are obtained by the system precisely to later carry out the respective signal processing. The algorithms implemented, which will be presented in detail in a future paper, are very complex and take into account many details that current systems do not.
The Figure 3 could serve as an outline for the paper. Consider adding some text to present the outline outline so readers can get situated. --> This has been solved.
Thank you. You are right.
In Section 2.4 System implementation consider clarifying the scope of the hardware early on: it just says "connect to a computer" - are there no existing commercial solutions? Is that for the experiments? Consider adding some introduction, it might not be obvious for the reader that it needed to process PC data and connect with a traffic monitoring system.
Thank you for improving the tables formatting. Consider using larger font, centering the text vertically and avoid smaller colored text, using multiple colors, especially low color contrast text (see Table 1). --> Thank you for your comments.
I am not sure what the Fig 14 is about. Is that a monitoring dashboard that displays real-time information from the PCB? Consider clarifying. --> Yes, it is. It is said in the paper that 'Fig. 14 shows the operation of our system at the described location and the supporting file 'Functioning 1', how it works in real time.' However, we have addes some more information to clarify it.
Thank you.
You mention that direction is important to capture - are you monitoring both directions? For the results section consider providing more details on the dataset: e.g., hours recorded, number of PC profiles captured, weather conditions encountered, hours of ground truth video and any other information that relates to the actual system performance like downtown. --> Of course the system is capable of monitoring both directions. In fact, Fuctioning 1 shows an example of this where a four-lane road (two in one direction and two in another) is monitored. We recommend watching the video.
On the conclusions, consider clarifying the contributions (lessons learned, system hardware, validation over a year...) and suggest future work. Again the arguments made in conclusion can be clarified, e.g., "The addition of functionalities to a sensor that nowadays is basically used only to measure the speed of vehicles and fine them if necessary." is difficult to justify given the related work. --> We assume that the readers of this journal have the necessary knowledge in ITS to understand what we mean. In addition, the introduction discuss about magnetic loops, CCTV ... therefore we believe that this is evident. However, we have made significant changes in this section.
We hope that these changes will satisfy your expectations and provide a higher quality to the manuscript.
Thank you very much again for your contribution. It has been a pleasure to have been in contact with you.
Reviewer 4 Report
The authors have done a good job addressing my initial concerns of ver. 1. A few items remain or were introduced in ver. 2. I suggest addressing these prior to publication.
I also look forward to the authors' noted follow-up paper on the machine learning aspects. This seems like the more significant aspect of their development.
Major issues:
line 116, "potential of this technology can be infinitely greater". I agree, much greater, or even stronger. "Infinitely" seems a bit excessive.
line 172, "It must discriminate up to 8 + 1 vehicle categories". Is there some reason you don't just say "9"? I'm not familiar with the German standard. Does the '+1" relate to trailers?' The terminology is probably fine, but some readers (including me) will wonder why you wrote it this way. Worth a clarifying comment or sentence. Or are you saying the "+ 1" is the "unknown" category? In my opinion, I'd just say something like "Here, '+ 1' corresponds to the 'unknown' category", something like that.
line 206, who is the reviewer in Table 1 (last line)? If appropriate, include a reference. If it is the authors, I suggest this is worth mentioning. Right now, these reviews don't mean much, since the basis is unclear. If there is no way to say, I suggest removing the review line from the table.
Minor issues:
line 101, "operation is straight forward" -> "operation is straightforward"
line 300, "printed circuit board" -> "PCB"
Author Response
The authors have done a good job addressing my initial concerns of ver. 1. A few items remain or were introduced in ver. 2. I suggest addressing these prior to publication. --> Thank you very much for the time spent on this second round of review. We honestly want to thank you for your comments and contributions. We fully agree with your suggestions. They have been very decisive in preparing a better version of the paper.
I also look forward to the authors' noted follow-up paper on the machine learning aspects. This seems like the more significant aspect of their development. --> We are also excited to publish this manuscript so that we can send the next one, which will be less informative and much more scientific and methodical.
Major issues:
line 116, "potential of this technology can be infinitely greater". I agree, much greater, or even stronger. "Infinitely" seems a bit excessive. --> Totally agree. In Spanish this word has no such connotation, but after reading your comment and reviewing that word in English, we consider that any of the options you propose is better. Thank you very much for detail.
line 172, "It must discriminate up to 8 + 1 vehicle categories". Is there some reason you don't just say "9"? I'm not familiar with the German standard. Does the '+1" relate to trailers?' The terminology is probably fine, but some readers (including me) will wonder why you wrote it this way. Worth a clarifying comment or sentence. Or are you saying the "+ 1" is the "unknown" category? In my opinion, I'd just say something like "Here, '+ 1' corresponds to the 'unknown' category", something like that. --> This standard is known worldwide as '8+1' because it discriminates between 8 vehicle categories (cars, cars with trailer, van, trucks, trucks with trailer, bus and articulated vehicles) and the 'extra category' of 'unknown'. Therefore, as 'unknown' is not a category as such, that terminology is used.
We tried in the second review to clarify this. However, in this second review, as you propose, we have rewriten this part to make it more clear to readers who are not familiar with the standard.
Now it is says: It includes 8 vehicle categories (motorcycles, cars, cars with trailer, van, trucks, trucks with trailer, bus and articulated vehicles) and the category 'unknown'. Hence the 8+1 nomenclature.
line 206, who is the reviewer in Table 1 (last line)? If appropriate, include a reference. If it is the authors, I suggest this is worth mentioning. Right now, these reviews don't mean much, since the basis is unclear. If there is no way to say, I suggest removing the review line from the table. --> You are right. This review was done by ourselves according to a template of technical specifications, which is detailed in one of the doctoral theses. However, it is true that this is not included in the manuscript and therefore it does not add much value. Therefore, after consideration, all the authors agree with you. So this last row in Table 1 has been removed.
Thank you very much for detail.
Minor issues:
line 101, "operation is straight forward" -> "operation is straightforward" --> Thank you very much for detail. This has been solved.
line 300, "printed circuit board" -> "PCB" --> Thank you very much for detail. This has been solved.
We hope that these changes will satisfy your expectations and provide a higher quality to the manuscript.
Thank you very much again for your contribution. It has been a pleasure to have been in contact with you.
This manuscript is a resubmission of an earlier submission. The following is a list of the peer review reports and author responses from that submission.
Round 1
Reviewer 1 Report
This paper investigates the problem of vehicle detection and classification using laser sensor, however, only laser choosing and configuration selecting are presented, it is not clear how to implement the vehicle detection and classification algorithm, which is believed to be important. In addition, the organization and presentation of the paper also needs to be improved.
Reviewer 2 Report
You have presented a good idea but it requires major improvement in terms of presentation and language used. Use precise writing instead of using long sentences. Try to capture reader's attention and interest while you present your method and results.
Connection of ideas require sequence which is missing from the paper. Some unnecessary detail has been provide while describing method that diverts reader's attention and interest.
Reviewer 3 Report
In general, this paper resembles a technical project report that details the development of a real time vehicle detection and classification. It is necessary to make several modifications to adapt the proposed work, in order to make it a scientific work for future submissions, with clear and original scientific contributions. The main recommendations are detailed as follows:
- Discussion regarding recent and related works must be improved. In the introduction, the limitation of some works is presented. However, the authors should extend such discussion (for instance, including a new section regarding related works), emphasizing limitations of other real time vehicle detection and classification works and original contributions of this paper. Some possible references for this section are presented as follows:
1) H. Liu, J. Ma, T. Xu, W. Yan, L. Ma and X. Zhang, "Vehicle Detection and Classification Using Distributed Fiber Optic Acoustic Sensing," in IEEE Transactions on Vehicular Technology, vol. 69, no. 2, pp. 1363-1374, Feb. 2020, doi: 10.1109/TVT.2019.2962334.
2) B. Sliwa, N. Piatkowski and C. Wietfeld, "The Channel as a Traffic Sensor: Vehicle Detection and Classification Based on Radio Fingerprinting," in IEEE Internet of Things Journal, vol. 7, no. 8, pp. 7392-7406, Aug. 2020, doi: 10.1109/JIOT.2020.2983207.
3) Song, H., Liang, H., Li, H. et al. Vision-based vehicle detection and counting system using deep learning in highway scenes. Eur. Transp. Res. Rev. 11, 51 (2019). https://doi.org/10.1186/s12544-019-0390-4
4) Zhang, F.; Li, C.; Yang, F. Vehicle Detection in Urban Traffic Surveillance Images Based on Convolutional Neural Networks with Feature Concatenation. Sensors 2019, 19, 594.
5) H. Wang, Y. Yu, Y. Cai, X. Chen, L. Chen and Q. Liu, "A Comparative Study of State-of-the-Art Deep Learning Algorithms for Vehicle Detection," in IEEE Intelligent Transportation Systems Magazine, vol. 11, no. 2, pp. 82-95, Summer 2019, doi: 10.1109/MITS.2019.2903518.
- The authors state that some related works are generally limited to speeds below 80 km/h. However, in the experiments, it is not clear whether the system was evaluated for speeds above 80km/h. This feature has not been included in the specifications too.
- All the calculations and mathematical representations must be presented in Subsection 2.3.
- Fig. 6 and 7 are irrelevant in the context of this work (scientific paper).
- The authors must detail the detection and classification algorithms used in this work.
- Comparisons with other vehicle detection and classification methods used in the literature must be included in this work.